# A Risk Decomposition Framework for Pre-Hoc Fine-Tuning Prediction

**Yuxiang Luo**[1]  **Chen Wang**[1]  **Nan Tang**[1]

## Abstract

The high cost of fine-tuning LLMs poses a significant economic barrier; pre-hoc performance prediction offers a critical solution to substantially reduce this expense. However, the theoretical limits of pre-hoc performance prediction remain unexplored. We formulate it as a stochastic estimation problem under information constraints, decomposing prediction risk into two components: an **intrinsic limit** (static data-model compatibility) and a **reducible optimization variance**. We prove that optimization variance admits a necessary lower bound on its decay rate, implying fundamental constraints on how quickly uncertainty dissipates, regardless of the predictor used. Based on these dynamics, we derive a budget-optimal probing principle and introduce a predictability phase diagram that organizes tasks into three distinct regimes: Static-Sufficient, Dynamic-Critical, and Noise-Dominant. Extensive experiments on synthetic and real-world benchmarks validate these theoretical regimes and demonstrate the efficiency of our probing strategy.

## 1. Introduction

Fine-tuning Large Language Models (LLMs) has become a central paradigm for adapting foundation models to downstream tasks (Zhang et al., 2025; Han et al., 2024; Zhu et al., 2022; Lin et al., 2025b). Despite its success, fine-tuning remains costly and uncertain: identical configurations can lead to substantially different outcomes due to interactions among pre-training priors, dataset characteristics, and stochastic optimization. In practice, large computational budgets are often spent only to yield limited gains, degraded performance, or catastrophic forgetting (Luo et al., 2025; Li et al., 2024), making trial-and-error fine-tuning increasingly impractical at scale.

[1]The Hong Kong University of Science and Technology (Guangzhou). Correspondence to: Nan Tang <nantang@hkust-gz.edu.cn>.

*Proceedings of the 43rd International Conference on Machine Learning*, Seoul, South Korea. PMLR 306, 2026. Copyright 2026 by the author(s).

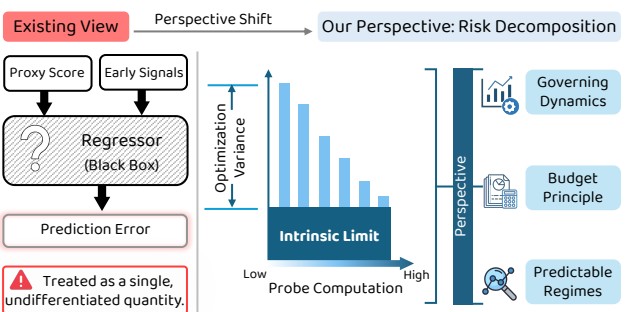

*Figure 1.* Perspective shift. We move from treating pre-hoc prediction as a black-box regression problem (Left) to a structural decomposition of risk into intrinsic limits and optimization variance (Right).

This motivates the problem of *pre-hoc fine-tuning prediction*: **estimating the final fine-tuning performance of a fine-tuning task before executing full training**. More concretely, given a pretrained model, a dataset, and an optimization algorithm, the goal is to predict the eventual outcome using only information available *prior to* or at the *very early stages* of optimization. Such predictions directly inform compute allocation decisions, including whether to continue training, which configurations to prioritize, and how much budget to invest.

To address this challenge, recent work has explored estimating fine-tuning outcomes before committing full training resources, a setting commonly referred to as *pre-hoc performance prediction* (Zeng et al., 2025b; Anugraha et al., 2024; Zeng et al., 2025a; Kuramoto & Suzuki, 2025). Empirical approaches, including proxy-based methods (Anugraha et al., 2024) and early-stage probing frameworks (Zhu et al., 2022), show that fine-tuning performance can be predicted to a non-trivial degree by combining static dataset statistics with short-horizon optimization signals. However, existing methods rely largely on heuristic designs and lack a principled framework for reasoning about *prediction error*, *uncertainty evolution*, and *resource allocation* (Zhu et al., 2022; Zeng et al., 2025a).

Most existing predictors operate as black-box regressors, treating optimization dynamics as features optimized for correlation with final outcomes (Anugraha et al., 2024). Within this view, prediction error is treated as a single undifferentiated quantity, and probing depth is handled as

a discrete hyperparameter (Kweon et al., 2025). In contrast, we adopt a structural perspective grounded in risk decomposition (Figure 1) (Kendall & Gal, 2017). We argue that unpredictability in fine-tuning arises from two distinct sources (Belkin et al., 2019; Guo et al., 2025): an intrinsic component determined by static data-model compatibility (the *Intrinsic Limit*) and a computation-dependent component induced by stochastic optimization (the *Optimization Variance*) (Mosbach et al., 2021). Under this view, probing resolves optimization-induced uncertainty rather than merely extracting features.

We formalize this perspective by modeling pre-hoc fine-tuning prediction as an estimation problem under progressively revealed information. Instead of treating probing as a fixed design choice, we characterize how prediction uncertainty evolves with allocated computation. We show that the Bayes-optimal prediction risk decomposes into a computation-independent intrinsic limit and a reducible optimization-induced variance (Walha et al., 2025). Moreover, without modeling full optimization dynamics, we derive a necessary lower-bound constraint on how fast optimization-induced uncertainty can decay with computation (Guo et al., 2025), governed by stochastic approximation dynamics (Zeng et al., 2025b; Lin et al., 2024). This implies that uncertainty cannot vanish arbitrarily fast and that pre-hoc prediction is inherently a problem of budget allocation with diminishing returns.

Our contributions are four-fold. First, we derive a formal risk decomposition separating intrinsic ambiguity from computation-dependent uncertainty. Second, we establish a theoretical lower bound on uncertainty decay, characterizing how quickly prediction uncertainty can be reduced. Third, we formulate pre-hoc prediction as a risk-cost tradeoff and derive a principled condition for budget-optimal probing. Finally, we synthesize intrinsic ambiguity and uncertainty dynamics into a predictability phase diagram that explains diverse empirical successes and failures of probing-based predictors.

## 2. Related Work

**Pre-hoc performance prediction.** Proxy-based methods (Anugraha et al., 2024; Wang et al., 2025) estimate performance using smaller models or data subsets, relying primarily on static correlations that often degrade under distribution shift (Zeng et al., 2025a). Trajectory-based approaches (Domhan et al., 2015; Klein et al., 2017) extrapolate final performance from early training statistics. While empirically effective, these methods are predominantly heuristic: probing depth is treated as a fixed design choice, and predictors are typically implemented as black-box regressors, offering limited insight into how uncertainty evolves with computation.

**Scaling laws and training dynamics.** Empirical scaling laws (Kaplan et al., 2020; Hoffmann et al., 2022) establish that expected model performance scales as a power-law function of compute, data size, and model capacity, with extensions to downstream fine-tuning and transfer learning (Isik et al., 2026; Schram et al., 2025; Barnett, 2024). However, this literature primarily characterizes the *first moment* of performance distributions. In parallel, training dynamics have been studied through critical learning periods, information bottleneck theory (Westphal et al., 2025; Saxe et al., 2018), and the structure of gradient noise (Sclocchi et al., 2023; Zeng et al., 2025b; Lin et al., 2024), suggesting that optimization outcomes are often determined early in training (Achille et al., 2019). These works are largely descriptive and do not provide a normative framework for allocating computation. In contrast, our approach focuses on the *second moment* of fine-tuning outcomes, offering a principled account of uncertainty decay and budget-aware prediction grounded in risk decomposition.

**Data-centric and cost-aware LLM pipelines.** A related but orthogonal line of work reduces the cost of LLM-powered systems through data, prompting, or pipeline design rather than predicting the final outcome of a candidate fine-tuning run. Examples include cost-effective in-context learning for entity resolution, weak-to-strong prompting for data transformation, natural-language data preparation, and LLM-based data-management pipelines (Fan et al., 2024; Li et al., 2025; Fan et al., 2025; Chen et al., 2023). These works motivate a broader view of compute-aware model development; our framework instead asks how much probe computation should be allocated before committing to full fine-tuning.

## 3. Problem Setup and Notation

We formalize pre-hoc fine-tuning prediction as the problem of **estimating a stochastic outcome under information constraints**. A fine-tuning *task instance* is defined as a tuple $\mathcal{T} = (M, D, \mathcal{A})$, where $M$ is the pre-trained foundation model, $D$ is the downstream dataset, and $\mathcal{A}$ denotes the stochastic optimization algorithm. Executing $\mathcal{T}$ yields a scalar performance metric $R \in \mathbb{R}$, which is treated as a random variable drawn from a task-conditional distribution $P(R \mid \mathcal{T})$ due to inherent randomness in optimization (e.g., initialization, data ordering and non-deterministic operations) (Gargiani et al., 2019).

**Predictors and task-conditional risk.** A predictor is a function $f : \mathcal{I} \to \mathbb{R}$ that maps an available information set to an estimate $\hat{R}$. For a *fixed task* $\mathcal{T}$, we define the *task-conditional prediction risk* at computation budget $c$ as:

$$\mathcal{L}_{\mathcal{T}}(c) \triangleq \mathbb{E}_{R|\mathcal{T}}\big[(f(\mathcal{I}_c) - R)^2\big]. \tag{1}$$

For any fixed information set $\mathcal{I}_c$, this risk is minimized by the Bayes-optimal predictor $f^*(\mathcal{I}_c) = \mathbb{E}[R \mid \mathcal{I}_c, \mathcal{T}]$, which serves as the reference object for our theoretical analysis in Section 4. All quantities derived in Sections 4–8, including the intrinsic limit and uncertainty decay rate, are defined *at the task level* through $\mathcal{L}_{\mathcal{T}}(c)$.

**Population risk.** When analyzing collections of tasks, we additionally consider the *population-level risk*, defined as the expectation of the task-conditional risk over a task distribution:

$$\mathcal{L}(c) \triangleq \mathbb{E}_{\mathcal{T}}[\mathcal{L}_{\mathcal{T}}(c)] = \mathbb{E}_{\mathcal{T}, R}\left[(f(\mathcal{I}_c) - R)^2\right]. \quad (2)$$

This quantity is used primarily for empirical aggregation and reporting, while the structural properties studied in this paper are governed by the task-conditional risk in Eq. (1).

**Information structure.** The information available to the predictor at computation budget $c$ is denoted by $\mathcal{I}_c = \{X_s, X_d^{(c)}\}$, where $X_s$ represents static, ex-ante information available at zero marginal cost (e.g., data–model compatibility signals), and $X_d^{(c)}$ denotes dynamic information revealed by executing the optimization algorithm up to budget $c$ (e.g., observed training trajectories). By construction, information is monotonic in computation: $\mathcal{I}_c \subseteq \mathcal{I}_{c'}$ for $c < c'$.

**Computation cost and budget constraint.** Acquiring information incurs a computational cost $C(c)$, which we assume to be strictly increasing with $C(0) = C_s$. For analytical tractability, we adopt a linear cost model $C(c) = C_s + \lambda c$, where $\lambda$ denotes the marginal unit cost of probing. Given a total budget $B$, the practitioner's objective is to choose an optimal stopping point $c^*$ that minimizes prediction risk subject to the cost constraint:

$$\min_{c \geq 0} \mathcal{L}_{\mathcal{T}}(c) \quad \text{s.t.} \quad C(c) \leq B. \quad (3)$$

This formulation makes explicit that all subsequent analysis concerns how *task-conditional uncertainty* decreases with computation, and how limited resources should be allocated to approach this limit efficiently.

## 4. Theoretical Framework: Risk Decomposition

In Section 3, we defined the prediction risk $\mathcal{L}(c)$ as a function of the computation budget $c$ through the information set $\mathcal{I}_c$. To understand the fundamental limits of pre-hoc prediction, we consider an ideal observer with access to the *asymptotic information set* $\mathcal{I}_\infty \triangleq \lim_{c \to \infty} \mathcal{I}_c$, corresponding to the complete optimization trajectory. This perspective allows us to characterize what aspects of prediction uncertainty are fundamentally irreducible and which can, in principle, be resolved by additional computation.

For the Bayes-optimal predictor $f^*(\mathcal{I}_c) = \mathbb{E}[R \mid \mathcal{I}_c]$, the prediction risk admits the following decomposition, which follows directly from the law of total variance (Kunievsky & Evans, 2026).

**Proposition 4.1** (Risk Decomposition)**.** *The Bayes-optimal prediction risk at computation budget $c$ can be written as*

$$\mathcal{L}(c) = \underbrace{\mathbb{E}[\text{Var}(R \mid \mathcal{I}_\infty)]}_{\mathcal{L}_{int}} + \underbrace{\mathbb{E}[\text{Var}(R \mid \mathcal{I}_c) - \text{Var}(R \mid \mathcal{I}_\infty)]}_{\mathcal{V}_{opt}(c)}.$$
$$(4)$$

*Here, $\mathcal{L}_{int}$ denotes the* intrinsic limit *of predictability—the residual uncertainty that remains even with full access to the optimization trajectory—while $\mathcal{V}_{opt}(c)$ denotes the* optimization variance, *i.e., the excess uncertainty due to observing only a finite prefix of the trajectory.*

The decomposition in Proposition 4.1 plays a purely organizational role: it separates prediction error into a computation-independent component and a computation-dependent component. The intrinsic term $\mathcal{L}_{int}$ represents an irreducible error floor arising from static data–model mismatch and inherent stochasticity in the task. By construction, it is invariant to the probing budget $c$ and cannot be reduced by allocating additional computation. The decomposition follows directly from the law of total variance; a formal proof is provided in Appendix A.

In contrast, the optimization variance $\mathcal{V}_{opt}(c)$ captures the portion of uncertainty that is, in principle, reducible through dynamic observation of the optimization process. Since the information set $\mathcal{I}_c$ grows monotonically with computation, $\mathcal{V}_{opt}(c)$ is non-negative and monotonically decreasing in $c$. Crucially, this is the *only* component of the prediction risk that can be affected by probing, making it the central object of interest in subsequent analysis.

This decomposition also admits a natural information-theoretic interpretation. Dynamic probing can reduce $\mathcal{V}_{opt}(c)$ only if the observed optimization trajectory contains information about the outcome $R$ that is not present in static priors. Formally, this requires the conditional mutual information $I(R; X_d^{(c)} \mid X_s)$ to be strictly positive. If the trajectory is fully determined by static information (i.e., $I(R; X_d^{(c)} \mid X_s) = 0$), then probing reveals no new information and cannot improve prediction accuracy. From this viewpoint, pre-hoc prediction is best understood as an *information revelation process*, where computation controls how rapidly reducible uncertainty $\mathcal{V}_{opt}(c)$ can be resolved.

## 5. Governing Dynamics: An Asymptotic Constraint on Uncertainty Decay

In Section 4, we identified the optimization variance $\mathcal{V}_{opt}(c)$ as the only component of prediction risk that can be reduced through additional computation. We now characterize a *fun-*

*damental asymptotic constraint* on how fast this reducible uncertainty can dissipate as computation increases. Our goal is not to model the full non-convex trajectory of large language model fine-tuning. Instead, we seek a conservative *rate-limiting envelope*: a necessary constraint on the decay of observable optimization-induced uncertainty once the optimization process enters a locally regular regime.

This distinction is important because fine-tuning dynamics need not be globally regular. The early trajectory may include transient behavior, abrupt loss changes, or task-specific adaptation phases that are not well described by a single asymptotic law. However, practical fine-tuning typically starts from a high-quality pre-trained solution, and informative gradient signals often emerge within a neighborhood of the initialization. In such locally stable regions, stochastic approximation theory provides a useful description of how gradient noise is gradually averaged out by the optimizer. We therefore analyze a regime in which the optimization dynamics are locally regular and the evaluation metric $R$ is locally Lipschitz with respect to the model parameters. The resulting statement should be interpreted as an asymptotic envelope for the rate-limited phase of uncertainty contraction, rather than a pointwise description of the entire training process.

Within this locally regular regime, the decay of optimization-induced uncertainty is constrained by the interaction between gradient noise, the optimizer's step-size schedule, and the sensitivity of the evaluation metric to parameter perturbations. This yields the following necessary asymptotic constraint.

**Proposition 5.1** (Asymptotic Uncertainty Decay Constraint)**.** *Consider a stochastic optimization process that admits a locally regular asymptotic regime in the sense described above. Then there exists a constant $K > 0$ and an exponent $\alpha > 0$ such that the optimization-induced uncertainty admits the asymptotic envelope*

$$\mathcal{V}_{opt}(c) \; \gtrsim \; K\,c^{-\alpha} \quad as\; c \to \infty, \qquad (5)$$

*where $\alpha$ is* an *effective decay rate jointly determined by the local optimization dynamics and the sensitivity of the evaluation metric. A formal derivation under classical stochastic approximation conditions is provided in Appendix C.*

*Proof Sketch.* The argument follows from the asymptotic behavior of stochastic optimization in a locally stable regime. Under standard stochastic approximation conditions, parameter uncertainty contracts at a rate governed by the decay of the step size and the structure of gradient noise. For example, for a representative polynomially decaying step-size schedule $\eta_c \propto c^{-\rho}$ with $\rho > 1/2$, the covariance of the parameter iterate contracts no faster than $\mathrm{Cov}(\theta_c) = \Omega(c^{-(2\rho-1)})$. When the evaluation metric $R(\theta)$

is locally smooth, the Delta method implies that fluctuations in $R$ inherit a corresponding polynomial decay. This yields a power-law envelope governing the observable contraction of optimization-induced uncertainty. We emphasize that this argument characterizes an asymptotic rate constraint in the locally regular regime, not a tight pointwise bound for every phase of a non-convex training trajectory. □

**Mechanism.** At a high level, this behavior reflects the accumulation of stochastic gradient noise and its gradual attenuation through decaying step sizes. Because noise is suppressed only polynomially in typical stochastic optimization regimes, uncertainty about the eventual optimization outcome cannot collapse arbitrarily fast once the contraction becomes rate-limited. The exponent $\alpha$ therefore acts as an effective descriptor of information revelation under a given task–optimizer pair, rather than a universal constant of the optimizer or the model family.

**Operational Interpretation.** The effective decay rate $\alpha$ characterizes how rapidly uncertainty relevant for pre-hoc prediction is revealed as computation increases. Large values of $\alpha$ correspond to rapid contraction of uncertainty, where shallow probing can already reveal most regime-relevant information (the *Static-Sufficient* regime). Smaller values of $\alpha$ indicate delayed or prolonged uncertainty contraction, requiring extended probing to resolve outcome variability (the *Dynamic-Critical* regime). This perspective explains why fixed-step probing strategies can perform inconsistently across tasks and motivates the budget-aware allocation strategy developed in the next section.

# 6. Budget-Optimal Probing Strategy

Building on the governing dynamics derived in Section 5, we now address the question of *how much probing is worth performing*. Because optimization-induced uncertainty decays with diminishing returns, pre-hoc prediction naturally gives rise to an **optimal stopping problem under computational cost**.

Rather than treating the probing budget as a hard constraint, we formulate probing as a risk-cost tradeoff. This perspective reveals that the optimal probing depth is governed by task-specific uncertainty dynamics.

## 6.1. Risk-Cost Tradeoff Formulation

Let $\mathcal{L}_{\mathcal{T}}(c)$ denote the task-conditional Bayes risk at computation $c$, and let $C(c)$ denote the associated cost. We consider the tradeoff objective

$$\min_{c \geq 0} \; \mathcal{L}_{\mathcal{T}}(c) + \gamma\,C(c), \qquad (6)$$

where $\gamma > 0$ converts computation into an equivalent risk penalty.

Using the decomposition from Proposition 4.1, we write

$$\mathcal{L}_{\mathcal{T}}(c) = \mathcal{L}_{\mathcal{T},int} + \mathcal{V}_{opt}(c), \tag{7}$$

where $\mathcal{L}_{\mathcal{T},int}$ denotes the irreducible intrinsic limit of the task. Motivated by Section 5, we adopt a canonical power-law envelope for the optimization-induced uncertainty:

$$\mathcal{V}_{opt}(c) \approx K\,c^{-\alpha}, \qquad \alpha > 0, \tag{8}$$

where $\alpha$ is interpreted as an effective decay rate.

## 6.2. Equilibrium Condition

**Theorem 6.1** (Budget-Optimal Probing Equilibrium). *Assume $\mathcal{V}_{opt}(c) = Kc^{-\alpha}$ and a differentiable, strictly increasing cost function $C(c)$. Then any interior optimum $c^\star$ of (6) satisfies*

$$\left|\frac{d\mathcal{V}_{opt}(c)}{dc}\right|_{c=c^\star} = \gamma\, C'(c^\star). \tag{9}$$

**Closed-Form Solution (Linear Cost).** For a linear cost model $C(c) = C_s + \lambda c$, the equilibrium condition yields

$$c^\star = \left(\frac{\alpha K}{\gamma \lambda}\right)^{\frac{1}{\alpha+1}}. \tag{10}$$

## 6.3. Offline Estimation of Governing Dynamics

We now describe a practical procedure for estimating the governing dynamics $(\alpha, K, \mathcal{L}_{\mathcal{T},int})$. This procedure is best viewed as an *offline calibration step* for analysis, benchmarking, and probe-budget design, rather than as an oracle available during deployment. The calibrated quantities are task-level descriptors of uncertainty dynamics: they summarize how information is revealed under a given task–optimizer pair, and are not intended to be recovered exactly for every deployment instance.

Given a set of probing depths $\{c_1, \ldots, c_m\}$, we run probes at these depths and compute a *surrogate uncertainty proxy* $\widehat{U}(c_i)$ that reflects dispersion across repeated runs or predictions at depth $c_i$. This proxy need not equal the true Bayes risk; it is only required to preserve the relative decay behavior of uncertainty across probing budgets. Operationally, a probe is *lightweight* when it is short relative to the cost of full fine-tuning, yet long enough to enter the stable interval where regime-relevant information has begun to emerge. Thus, lightweightness is not tied to a universal number of steps, but to the risk.

We then jointly fit the model

$$\widehat{U}(c) \approx \mathcal{L}_{\mathcal{T},int} + Kc^{-\alpha} \tag{11}$$

via nonlinear or log–log regression, avoiding sequential plug-in bias. The resulting $\widehat{\alpha}$ is interpreted as an effective

---

**Algorithm 1** Offline Calibration of Budget-Optimal Probing

**Require:** Pretrained model $M$, dataset $D$, probing depths $\{c_1, \ldots, c_m\}$, tradeoff parameter $\gamma$, unit cost $\lambda$
**Ensure:** Estimated optimal probing depth $\widehat{c}^\star$
 1: Run lightweight probes at depths $\{c_1, \ldots, c_m\}$ and compute uncertainty proxies $\widehat{U}(c_i)$
 2: Jointly fit $(\widehat{\alpha}, \widehat{K}, \widehat{\mathcal{L}}_{\mathcal{T},int})$ from $\{\widehat{U}(c_i)\}$
 3: Compute equilibrium probing depth
 4: $\quad \widehat{c}^\star \leftarrow \left(\frac{\widehat{\alpha}\,\widehat{K}}{\gamma\,\lambda}\right)^{\frac{1}{\widehat{\alpha}+1}}$
 5: **return** $\widehat{c}^\star$

---

descriptor of the task's uncertainty dynamics, while $\widehat{\mathcal{L}}_{\mathcal{T},int}$ captures the empirical floor approached by the surrogate uncertainty proxy. In downstream use, these fitted quantities support coarse regime-level decisions and probe-budget selection, rather than exact latent-parameter recovery.

**Remarks.** Algorithm 1 avoids reliance on observing the final fine-tuning outcome and does not require access to complete training trajectories. The fitted decay rate $\widehat{\alpha}$ should be interpreted as a task-level, optimization-dependent descriptor rather than a universal constant. Repeated probes are used in offline analysis to reduce noise in estimating the surrogate uncertainty curve; they are not part of the deployment-time protocol. In practice, coarse fits over a small number of probing depths are sufficient to recover stable regime-level behavior, as demonstrated in Section 8 and the sensitivity analysis in Appendix F.

# 7. Phase Diagram and Predictability Regimes

The framework developed in Sections 4–6 implies that fine-tuning predictability cannot be captured by a single scalar notion of difficulty. Instead, predictability emerges from the interaction between two distinct mechanisms: the *intrinsic ambiguity* of the task, captured by $\mathcal{L}_{int}$, and the *temporal dynamics of information revelation*, captured by the effective decay rate $\alpha$.

In this section, we synthesize these factors into a **predictability phase diagram**. Rather than serving as a rigid categorization of tasks, this diagram provides a mechanism-based view of why certain tasks are predictable early, others only after extended computation, and some not at all.

## 7.1. Phase Space of Predictability

We represent each fine-tuning task $\mathcal{T}$ as a point in a two-dimensional phase space spanned by the governing quantities introduced earlier.

**Intrinsic limit ($\mathcal{L}_{int}$).** The horizontal axis corresponds to the irreducible prediction floor. Large values of $\mathcal{L}_{int}$ indi-

cate fundamental data–model mismatch or inherent stochasticity in the evaluation metric, which bounds the achievable prediction accuracy even under full information. Tasks with noisy labels, ambiguous supervision, or weak alignment between the evaluation metric and model behavior naturally exhibit elevated intrinsic limits.

**Effective decay rate ($\alpha$).** The vertical axis captures the speed at which optimization-induced uncertainty can be resolved. Small values of $\alpha$ correspond to slow or delayed information revelation, while larger values indicate that uncertainty collapses rapidly once informative gradients are available. This quantity reflects properties of the optimization trajectory itself, rather than static dataset characteristics.

Together, these axes define a landscape of predictability that is independent of any specific predictor architecture or probing heuristic. While probing is one operational means of revealing information, the phase diagram characterizes properties of the task–optimization pair itself.

## 7.2. Regimes of Information Revelation

The regimes described below correspond to qualitatively different behaviors of the risk–cost tradeoff derived in Section 6. Formally, they arise from distinct solution structures of the equilibrium condition in Theorem 6.1 as the task parameters $(\mathcal{L}_{int}, \alpha)$ vary.

**Regime I: Static-Sufficient (Bias-Dominant).** This regime arises when uncertainty collapses rapidly or when the intrinsic limit dominates the total risk. Here, the final outcome is effectively determined by coarse properties of the data–model pair, and dynamic observation provides little additional information beyond what is already encoded in static descriptors.

In practice, many surface-level adaptation tasks fall into this regime. For example, sentiment classification benchmarks such as SST-2, as well as several GLUE-style classification tasks, consistently exhibit large decay exponents $\alpha$ and reach their predictability saturation with minimal probing. In such cases, static priors alone nearly saturate achievable prediction accuracy, and extended trajectory observation yields diminishing returns.

**Regime II: Dynamic-Critical (Variance-Dominant).** This regime captures tasks with low intrinsic ambiguity but slow or delayed information revelation. Although accurate prediction is possible in principle, the outcome depends sensitively on the realization of the optimization trajectory.

Extended plateaus, saddle-point-dominated dynamics, and delayed generalization phenomena (e.g., grokking (Power et al., 2022)) naturally fall within this regime. Concrete examples include arithmetic and multi-step reasoning benchmarks such as GSM8K, as well as complex code generation tasks, where early-stage training dynamics carry essential predictive information. Within our framework, such tasks are characterized by small effective decay rates $\alpha$, leading to large optimal probing depths $c^*$.

**Regime III: Noise-Dominant (Intrinsic-Limited).** This regime arises when irreducible ambiguity dominates prediction error. Even with extensive probing, the total risk remains high due to a large intrinsic limit.

In this setting, additional probing yields diminishing practical benefits. The predictor is effectively modeling stochastic noise rather than a stable signal. Tasks with substantial label noise, extreme domain shift, or severe model underspecification typically exhibit this behavior, and pre-hoc prediction is fundamentally constrained.

## 7.3. Implications for Empirical Predictability

The phase diagram offers a unifying explanation for seemingly contradictory empirical observations. For instance, shallow probing strategies often succeed on benchmarks such as SST-2 not because the predictors are necessarily more expressive, but because these tasks reside in the static-sufficient regime, where uncertainty collapses rapidly. In contrast, the same strategies tend to fail on reasoning-intensive benchmarks such as GSM8K, not simply due to methodological shortcomings, but because such tasks lie in the dynamic-critical regime, where information revelation is intrinsically delayed.

This perspective also clarifies how different sources of predictive signal should be used. Static proxies, such as dataset statistics, reference-model perplexity, or data–model compatibility scores, are most informative when the dominant uncertainty is already encoded in the task before optimization begins. Early-trajectory signals, such as loss decay, gradient stability, or short-horizon validation behavior, become more valuable when reducible optimization variance remains substantial after static information is observed. The phase diagram therefore provides a way to interpret these signals not as competing alternatives, but as complementary views of different components of prediction risk.

Practically, this suggests a regime-aware use of pre-hoc predictors. For static-sufficient tasks, additional probing may provide little benefit beyond static descriptors, and shallow or even zero-probe predictors may be adequate. For dynamic-critical tasks, early optimization trajectories carry essential information, and reliable prediction requires allocating enough probe budget for the relevant dynamics to emerge. For noise-dominant tasks, neither richer static features nor deeper probing can fully overcome the intrinsic floor, so prediction scores should be interpreted with higher uncertainty.

From this perspective, the failure of a pre-hoc predictor should be interpreted not simply as a flaw of the predictor itself, but as a *regime mismatch* between the estimator's design, the allocated probing budget, and the task's governing dynamics. The practical role of the phase diagram is thus to guide when static information is sufficient, when dynamic probing is worth its cost, and when additional computation is unlikely to materially change the prediction decision.

This view also connects to data-quality and data-efficiency methods that treat dataset-level or early-training signals as actionable evidence, including early-loss-based mislabel detection, cost-effective missing-value imputation, coreset selection, and selective data acquisition (Deng et al., 2024; Chai et al., 2025; 2023; 2022). In our setting, analogous signals are not used to directly clean or acquire data, but to distinguish reducible optimization variance from intrinsic ambiguity.

## 8. Empirical Validation

Our theoretical framework makes *structural* predictions about fine-tuning predictability, rather than claims about the absolute accuracy of any specific predictor. In particular, it implies that: (i) optimization-induced uncertainty cannot dissipate arbitrarily fast and is governed by a task-dependent decay rate $\alpha$ (Section 5); (ii) fine-tuning tasks organize into distinct predictability regimes characterized by their effective intrinsic floor and uncertainty decay behavior (Section 8.3); and (iii) optimal probing follows a regime-dependent stopping structure rather than a fixed heuristic budget (Section 6).

In this section, we empirically validate these *structural predictions* using controlled experiments and large-scale fine-tuning benchmarks. All empirical quantities reported here are interpreted as *observable uncertainty proxies* that reflect optimization-induced variability, rather than direct estimates of the theoretical prediction risk. Detailed estimation procedures and justifications are provided in Appendix F.

### 8.1. Experimental Setup and Estimation Protocols

We adopt a unified probing protocol to empirically instantiate the key qualitative behaviors predicted by our framework, including uncertainty decay, regime structure, and diminishing returns of probing. The purpose of this protocol is structural validation: we use controlled repetitions to estimate observable uncertainty curves and to analyze how these curves organize tasks into predictability regimes. This differs from deployment-time use, where the calibrated regime structure serves as guidance for probe-budget selection and decision support rather than requiring repeated estimation of all latent quantities.

**Tasks and regime coverage.** We evaluate a diverse collection of fine-tuning tasks spanning different objectives and data conditions, selected to cover all three predictability regimes introduced in Section 8.3: *Static-Sufficient*, *Dynamic-Critical*, and *Noise-Dominant*. These include, for example, sentiment classification, arithmetic reasoning, and tasks with ambiguous or noisy supervision. Detailed experimental configurations are deferred to Appendix F.1.

**Probing axis and repeated runs.** We define the probing computation axis $c$ as the number of probe optimization steps, which serves as a consistent proxy for computational cost. For each task and each probing depth $c \in \mathcal{C}$, we perform $N = 1,500$ independent fine-tuning runs with different random seeds, yielding an empirical distribution of realized final outcomes. This repeated-run protocol is used to make the uncertainty decay behavior identifiable under controlled offline conditions. It is not intended as a deployment-time requirement: in deployment, the framework uses offline-calibrated descriptors and regime-level trends to guide how much probing is worth performing.

**Observable uncertainty proxies.** At each probing depth $c$, we compute an observable uncertainty proxy that captures run-to-run variability induced by stochastic optimization. This quantity does *not* estimate the theoretical prediction risk $L(c)$ defined in Section 3, nor does it assume access to asymptotically complete information. Instead, it serves as a measurable surrogate that preserves the relative uncertainty decay behavior analyzed in Sections 5–6. Accordingly, the empirical estimates of $\alpha$ and $\mathcal{L}_{\mathcal{T},int}$ should be interpreted as descriptors of the surrogate uncertainty curve, rather than exact online quantities to be recovered for every deployment instance. Formal definitions, normalization procedures, and fitting protocols are provided in Appendix F.

### 8.2. Decomposability and Power-Law Uncertainty Decay

We first examine the empirical behavior of observable uncertainty decay predicted by Proposition 5.1, which establishes a power-law constraint on how fast optimization-induced uncertainty can dissipate under stable training dynamics.

Figure 2 reports the population-level behavior of normalized observable uncertainty aggregated by regime. Across all regimes, the log–log relationship between uncertainty and probing depth exhibits an approximately linear trend within the fitted range, consistent with the rate-limiting power-law behavior predicted by theory.

Importantly, the empirically observed decay rates differ systematically across regimes. In *Static-Sufficient* regimes, uncertainty contracts rapidly, indicating that dynamic probing information is exhausted almost immediately. This behavior

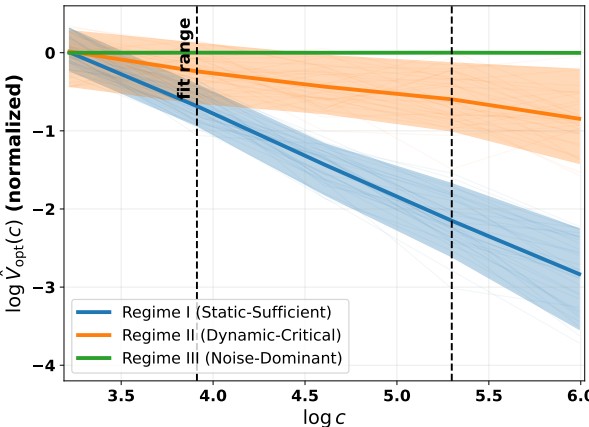

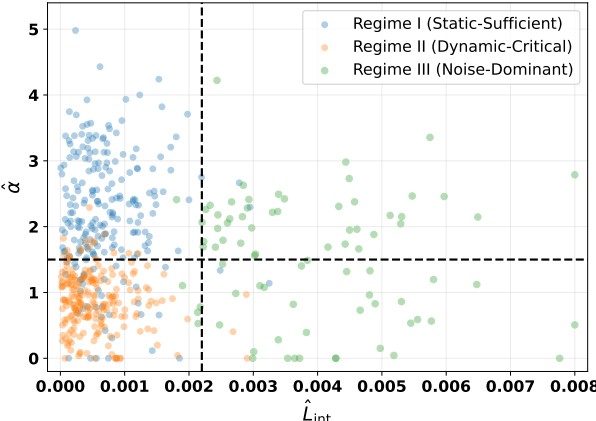

*Figure 2.* Population-level uncertainty decay across regimes. The figure shows normalized observable uncertainty as a function of probing depth $c$ in log-log scale, aggregated by regime. Solid curves represent regime-wise means across tasks, with shaded regions indicating bootstrap confidence intervals. Vertical dashed lines denote the stable probing interval used for fitting.

*Figure 3.* Empirical phase diagram of fine-tuning tasks. Each point corresponds to a task, positioned by its estimated effective intrinsic floor and empirically observed decay exponent. Colors indicate regime assignments. Dashed reference lines are shown for interpretability and reflect regime-level structure rather than learned decision boundaries.

reflects that probing is largely unnecessary, as outcomes are already well-determined by static task characteristics.

In contrast, *Dynamic-Critical* regimes exhibit substantially slower decay, with uncertainty persisting over a wide range of probing depths. This pattern indicates that early-stage optimization dynamics contain essential predictive information, making probing critical under practical budgets.

Finally, in *Noise-Dominant* regimes, observable uncertainty remains dominated by an effective intrinsic floor, indicating that prediction difficulty arises primarily from irreducible ambiguity rather than insufficient probing.

Together, these results demonstrate that the decay exponent $\alpha$ captures a meaningful notion of *dynamic hardness* and provides an empirical counterpart to the governing uncertainty dynamics characterized in Section 5.

### 8.3. Phase Diagram and Regime Assignment

We next examine how fine-tuning tasks are organized in the two-dimensional space induced by their effective intrinsic floor and uncertainty decay rate. This empirical phase diagram provides a concrete instantiation of the regime taxonomy introduced in Section 8.3.

Figure 3 plots all tasks in this reduced space. A clear structural separation emerges. *Static-Sufficient* tasks concentrate in the region characterized by low intrinsic ambiguity and rapid uncertainty decay, indicating that outcomes are largely determined by static factors. In contrast, *Dynamic-Critical* tasks occupy regions with similarly low intrinsic floors but markedly slower decay, reflecting the decisive role of training dynamics in predictability.

*Noise-Dominant* tasks are primarily separated along the intrinsic-floor axis, spreading toward higher ambiguity across a wide range of decay rates. This pattern highlights that a large intrinsic floor, rather than the decay rate alone, governs predictability in this regime.

Overall, the empirical phase diagram aligns closely with the theoretical regime definitions and demonstrates that the proposed descriptors provide a meaningful low-dimensional representation of fine-tuning behavior.

### 8.4. Efficiency Frontier and Optimal Probing

We next study how probing budgets should be allocated in practice. Building on the regime-dependent uncertainty decay identified above, we analyze the marginal benefit of additional probing and its implications for optimal stopping under limited resources.

Figure 4 visualizes the marginal gain of probing, defined as the reduction in normalized observable uncertainty achieved by an incremental increase in probing depth (see Appendix F.6 for a precise definition). Consistent with the analysis in Section 6, marginal gains decay monotonically with increasing $c$ across all regimes.

However, the rate at which marginal gains decay differs substantially. In *Static-Sufficient* regimes, marginal gains vanish almost immediately, indicating that additional probing is unnecessary. In *Dynamic-Critical* regimes, marginal gains persist over a wide range of budgets, making extended probing cost-effective. In *Noise-Dominant* regimes, marginal gains remain negligible due to the dominance of the effective intrinsic floor.

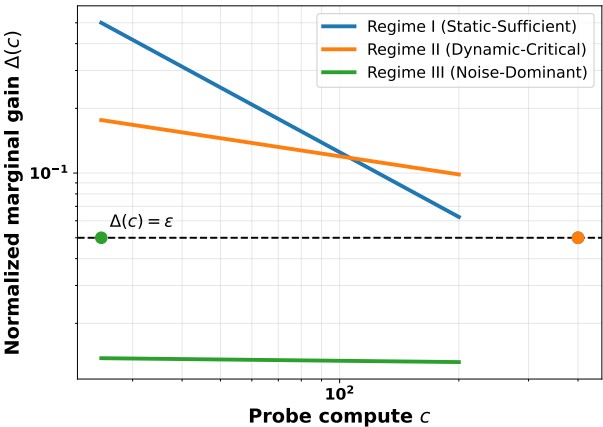

*Figure 4.* Regime-dependent marginal gain of probing. The figure plots the normalized marginal reduction in observable uncertainty $\Delta(c)$ as a function of probing depth $c$ for different regimes. All curves exhibit diminishing returns, but at markedly different rates.

These patterns empirically validate the regime-aware stopping principles derived in Theorem 6.1. They also suggest that the practical value of probing lies in identifying the point at which additional computation mainly refines the uncertainty estimate without materially changing the regime-level decision. Appendix F.7 provides a complementary probe-budget sensitivity analysis showing how regime agreement and uncertainty estimates evolve with increasing probe depth.

### 8.5. Regime Mismatch and Empirical Failures

We examine empirical failure cases of probing-based predictors. Rather than attributing failures to estimator design, our framework explains them through *structural regime mismatch* between task dynamics and applied probing strategy.

Probing is informative only when dynamic uncertainty is both present and reducible. In *Static-Sufficient* regimes, outcomes are largely determined by static task characteristics, so dynamic uncertainty is exhausted immediately. In *Noise-Dominant* regimes, prediction error is dominated by a large intrinsic risk floor, making additional probing ineffective.

**Case I: Static-Sufficient Regime.** Sentiment classification tasks such as SST-2 provide a representative example. Probing-based predictors reach near-saturated performance after only a few steps, with consistently large estimated decay rates, indicating rapid information revelation.

As a result, deeper probing yields diminishing returns and may introduce unnecessary variance from stochastic optimization. In this regime, apparent probing failures should therefore be interpreted as applying dynamic probing where static descriptors already suffice.

**Case II: Dynamic-Critical Regime.** Arithmetic reasoning benchmarks such as GSM8K exhibit markedly different behavior. Although the intrinsic ambiguity $\mathcal{L}_{int}$ remains low, the empirically observed decay rate is small, indicating slow and delayed information revelation. Optimization trajectories often exhibit extended plateaus, leading to prolonged predictive uncertainty.

Here, shallow probing underestimates uncertainty and produces unstable predictions. Failures arise not because probing is flawed, but because the probing depth is insufficient relative to the task's governing dynamics. This behavior is characteristic of the *Dynamic-Critical* regime, where reliable prediction emerges only after substantial computation.

Additional ablation results and qualitative analyses are provided in Appendix F.10 and Appendix F.11.

## 9. Conclusion and Limitations

This work studies pre-hoc fine-tuning prediction from a structural perspective by treating it as a problem of uncertainty resolution under computational constraints. We characterize the limits of predictability by decomposing prediction risk into an intrinsic component and a computation-dependent optimization variance, establishing a necessary constraint on how fast this uncertainty can dissipate, and organizing fine-tuning tasks into a regime-dependent phase space. Together, these results explain when static information is already sufficient, when dynamic probing reveals essential additional signal, and when prediction is limited by an intrinsic uncertainty floor. This provides a principled basis for budget-aware probing decisions beyond heuristic step-count rules, and offers a way to interpret existing static-proxy and early-trajectory predictors through the components of prediction risk they primarily capture.

Our analysis avoids modeling the full non-convex optimization dynamics of large language models. In particular, we do not claim that fine-tuning trajectories universally follow a specific decay law, nor that the decay exponent $\alpha$ corresponds to a fixed physical constant. Instead, $\alpha$ should be interpreted as an effective descriptor of how quickly uncertainty is revealed within a locally regular, rate-limited regime under a given task–optimizer pair. Similarly, the intrinsic floor and decay parameters are best understood as structural descriptors for offline calibration and regime-level decision support, rather than quantities that must be recovered exactly in every deployment instance. Task regimes may also shift under different hyperparameters, model families, or training protocols. These limitations reflect the intended scope of the framework and point to future directions where more detailed dynamical modeling, adaptive stopping rules, and integration with practical predictor architectures may further improve pre-hoc fine-tuning decisions.

## Acknowledgements

This work is supported by Guangdong provincial project (Project No. 2023CX10X008).

## Impact Statement

This work aims to advance the theoretical understanding of pre-hoc fine-tuning prediction through a risk decomposition framework. A potential positive impact is improved efficiency in model development: by clarifying when early probing is informative, redundant, or structurally limited, the framework can help practitioners make more informed decisions about whether and how long to fine-tune a model. This may reduce unnecessary computational expenditure, lower experimentation cost, and support more systematic allocation of fine-tuning resources.

At the same time, pre-hoc prediction should not be interpreted as a replacement for careful downstream evaluation. A potential risk is that practitioners may over-rely on predicted performance or regime assignments when deciding whether to continue training, especially in settings where evaluation data are noisy, incomplete, or misaligned with deployment objectives. Such misuse could prematurely discard useful datasets or configurations, or reinforce biases present in the surrogate signals used for probing. Our framework is therefore intended as a decision-support tool rather than an automatic acceptance or rejection mechanism.

The proposed analysis does not introduce new model architectures, training objectives, or deployed systems, and does not directly change model behavior. Nevertheless, when used in practical fine-tuning pipelines, we recommend reporting the probing budget, uncertainty estimates, task coverage, and limitations of the calibration protocol. These practices can help ensure that compute-saving decisions remain transparent, auditable, and appropriately qualified.

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

# A. Proofs for Risk Decomposition

This appendix provides the formal proofs and supporting results for the risk decomposition framework introduced in Section 4. We rigorously establish how the prediction risk decomposes into an irreducible intrinsic component and a reducible optimization-induced component under increasing information.

## A.1. Formal Setup and Bayes-Optimal Prediction

Let $R \in \mathbb{R}$ denote the random outcome of a fine-tuning task, and let $\{\mathcal{I}_c\}_{c \geq 0}$ denote a filtration representing the accumulated information available to the predictor as a function of computation $c$. By construction,

$$\mathcal{I}_c \subseteq \mathcal{I}_{c'} \quad \text{for all } c < c'. \tag{12}$$

For any fixed information set $\mathcal{I}_c$, consider predictors $f : \mathcal{I}_c \to \mathbb{R}$ evaluated under the mean squared error (MSE) risk

$$\mathcal{L}(f; c) \triangleq \mathbb{E}\big[(f(\mathcal{I}_c) - R)^2\big]. \tag{13}$$

It is a standard result in statistical decision theory that the Bayes-optimal predictor minimizing $\mathcal{L}(f; c)$ is given by the conditional expectation

$$f^*(\mathcal{I}_c) = \mathbb{E}[R \mid \mathcal{I}_c], \tag{14}$$

and that the corresponding minimal risk is

$$\mathcal{L}(c) = \mathcal{L}(f^*; c) = \mathbb{E}\big[(R - \mathbb{E}[R \mid \mathcal{I}_c])^2\big] = \mathbb{E}[\mathrm{Var}(R \mid \mathcal{I}_c)]. \tag{15}$$

This identity serves as the starting point for all subsequent derivations.

## A.2. Proof of Proposition 4.1

We now prove the risk decomposition stated in Proposition 4.1.

**Restatement.** Let $\mathcal{I}_\infty = \lim_{c \to \infty} \mathcal{I}_c$ denote the information set corresponding to the complete optimization trajectory. Then the Bayes risk at computation $c$ admits the decomposition

$$\mathcal{L}(c) = \underbrace{\mathbb{E}[\mathrm{Var}(R \mid \mathcal{I}_\infty)]}_{\mathcal{L}_{int}} + \underbrace{\mathbb{E}[\mathrm{Var}(R \mid \mathcal{I}_c) - \mathrm{Var}(R \mid \mathcal{I}_\infty)]}_{\mathcal{V}_{opt}(c)}. \tag{16}$$

**Proof.** Starting from the expression for the Bayes risk,

$$\mathcal{L}(c) = \mathbb{E}[\mathrm{Var}(R \mid \mathcal{I}_c)], \tag{17}$$

we add and subtract $\mathbb{E}[\mathrm{Var}(R \mid \mathcal{I}_\infty)]$ to obtain

$$\mathcal{L}(c) = \mathbb{E}[\mathrm{Var}(R \mid \mathcal{I}_\infty)] + \mathbb{E}[\mathrm{Var}(R \mid \mathcal{I}_c) - \mathrm{Var}(R \mid \mathcal{I}_\infty)]. \tag{18}$$

The first term depends only on the full information set $\mathcal{I}_\infty$ and is therefore independent of $c$. We define this term as the intrinsic limit

$$\mathcal{L}_{int} \triangleq \mathbb{E}[\mathrm{Var}(R \mid \mathcal{I}_\infty)]. \tag{19}$$

The second term captures the excess uncertainty due to observing only a finite prefix of the optimization trajectory and is defined as

$$\mathcal{V}_{opt}(c) \triangleq \mathbb{E}[\mathrm{Var}(R \mid \mathcal{I}_c) - \mathrm{Var}(R \mid \mathcal{I}_\infty)]. \tag{20}$$

This yields the stated decomposition. $\square$

## A.3. Properties of the Optimization Variance Term

We now establish basic structural properties of $\mathcal{V}_{opt}(c)$.

### A.3.1. Non-negativity

**Claim.**
$$\mathcal{V}_{opt}(c) \geq 0 \quad \text{for all } c \geq 0. \tag{21}$$

**Proof.** Since $\mathcal{I}_c \subseteq \mathcal{I}_\infty$, conditioning on $\mathcal{I}_\infty$ provides at least as much information as conditioning on $\mathcal{I}_c$. A standard property of conditional variance implies

$$\text{Var}(R \mid \mathcal{I}_c) \geq \text{Var}(R \mid \mathcal{I}_\infty) \quad \text{almost surely.} \tag{22}$$

Taking expectations on both sides yields $\mathcal{V}_{opt}(c) \geq 0$. $\qquad\square$

### A.3.2. Monotonicity with Respect to Computation

**Claim.** If $c < c'$, then
$$\mathcal{V}_{opt}(c) \geq \mathcal{V}_{opt}(c'). \tag{23}$$

**Proof.** By the filtration property, $\mathcal{I}_c \subseteq \mathcal{I}_{c'} \subseteq \mathcal{I}_\infty$. Applying the monotonicity of conditional variance,

$$\text{Var}(R \mid \mathcal{I}_c) \geq \text{Var}(R \mid \mathcal{I}_{c'}) \geq \text{Var}(R \mid \mathcal{I}_\infty). \tag{24}$$

Subtracting $\text{Var}(R \mid \mathcal{I}_\infty)$ and taking expectations yields

$$\mathcal{V}_{opt}(c) \geq \mathcal{V}_{opt}(c'). \tag{25}$$

$$\square$$

### A.3.3. Asymptotic Vanishing

**Claim.**
$$\lim_{c \to \infty} \mathcal{V}_{opt}(c) = 0. \tag{26}$$

**Proof.** By definition, $\mathcal{I}_c \uparrow \mathcal{I}_\infty$ as $c \to \infty$. Under standard regularity assumptions ensuring convergence of conditional expectations (e.g., dominated convergence),

$$\lim_{c \to \infty} \text{Var}(R \mid \mathcal{I}_c) = \text{Var}(R \mid \mathcal{I}_\infty) \quad \text{almost surely.} \tag{27}$$

Taking expectations yields the result. $\qquad\square$

### A.4. Interpretation: Floor and Gap

The results above formally justify the interpretation adopted in the main text. The intrinsic limit $\mathcal{L}_{int}$ represents an irreducible error floor determined by the task itself and remains invariant to computation. In contrast, the optimization variance $\mathcal{V}_{opt}(c)$ represents a reducible uncertainty gap that shrinks monotonically as additional information from the optimization trajectory is revealed. This structural separation underpins the governing dynamics analyzed in Section 5 and the budget-optimal probing strategy developed in Section 6.

## B. Information-Theoretic Interpretation and Properties

This appendix complements Section 4 by formalizing the information-theoretic perspective behind static–dynamic complementarity and the "information revelation" viewpoint. Throughout, let $\mathcal{I}_c = \{X_s, X_d^{(c)}\}$ denote the accumulated information at computation $c$, and let $\mathcal{I}_\infty = \{X_s, X_d^{(\infty)}\}$ denote the asymptotic information set.

### B.1. Conditional Mutual Information as a Criterion for Complementarity

We first relate the usefulness of dynamic probes to the conditional mutual information

$$I(R; X_d^{(c)} \mid X_s). \tag{28}$$

Intuitively, this quantity measures the *non-redundant* information about the outcome $R$ that is revealed by observing the optimization trajectory prefix beyond what is already contained in static priors.

**Lemma B.1** (Redundancy implies no gain). *If $I(R; X_d^{(c)} \mid X_s) = 0$, then conditioning on $X_d^{(c)}$ does not change the posterior of $R$ given $X_s$, i.e.,*

$$p(R \mid X_s, X_d^{(c)}) = p(R \mid X_s) \quad a.s., \tag{29}$$

*and consequently*

$$\mathbb{E}\Big[\mathrm{Var}(R \mid X_s, X_d^{(c)})\Big] = \mathbb{E}[\mathrm{Var}(R \mid X_s)]. \tag{30}$$

**Proof.** The condition $I(R; X_d^{(c)} \mid X_s) = 0$ is equivalent to conditional independence $R \perp\!\!\!\perp X_d^{(c)} \mid X_s$, which implies $p(R \mid X_s, X_d^{(c)}) = p(R \mid X_s)$ almost surely. Taking conditional variances under the same conditional distribution yields $\mathrm{Var}(R \mid X_s, X_d^{(c)}) = \mathrm{Var}(R \mid X_s)$ almost surely, and the expectation identity follows. $\square$

**Corollary B.2** (Strict variance reduction requires positive conditional MI). *If $\mathbb{E}[\mathrm{Var}(R \mid X_s, X_d^{(c)})] < \mathbb{E}[\mathrm{Var}(R \mid X_s)]$, then $I(R; X_d^{(c)} \mid X_s) > 0$.*

**Proof.** By Lemma B.1, $I(R; X_d^{(c)} \mid X_s) = 0$ would imply equality of the two expected conditional variances, contradicting strict reduction. $\square$

## B.2. Monotone Information Revelation Along the Filtration

Because $\mathcal{I}_c$ grows with $c$ (as a filtration), information revealed by dynamic probes must be monotone in computation.

**Lemma B.3** (Monotonicity of conditional mutual information). *If $c < c'$, then*

$$I(R; X_d^{(c)} \mid X_s) \leq I(R; X_d^{(c')} \mid X_s). \tag{31}$$

**Proof.** Since $X_d^{(c)}$ is a measurable function of $X_d^{(c')}$ (the latter contains the former as a prefix), we have the Markov chain

$$R \to (X_s, X_d^{(c')}) \to (X_s, X_d^{(c)}),$$

and the claim follows from the data-processing inequality for conditional mutual information. $\square$

This lemma formalizes the "information revelation" interpretation: as computation increases, the dynamic observation can only add (never remove) information about the outcome, and can only decrease the Bayes risk $\mathcal{L}(c) = \mathbb{E}[\mathrm{Var}(R \mid \mathcal{I}_c)]$.

## B.3. From Mutual Information to Variance Reduction

Section 4 motivates complementarity through conditional mutual information. Here we provide a precise quantitative relationship between mutual information and variance reduction. In full generality, mutual information is defined in terms of (differential) entropies:

$$I(R; X_d^{(c)} \mid X_s) = h(R \mid X_s) - h(R \mid X_s, X_d^{(c)}), \tag{32}$$

where $h(\cdot)$ denotes differential entropy (for continuous $R$) and the discrete entropy analog applies if $R$ is discrete.

A direct link to conditional variances emerges under a standard (and explicitly stated) Gaussianity condition, which is commonly adopted when analyzing posterior contraction and stochastic approximation in local regimes.

**Proposition B.4** (Gaussian posterior contraction identity). *Assume that for almost every realization of $(X_s, X_d^{(c)})$, the conditional distribution $R \mid (X_s, X_d^{(c)})$ is Gaussian with variance $\sigma_c^2(X_s, X_d^{(c)})$, and $R \mid X_s$ is Gaussian with variance $\sigma_0^2(X_s)$. Then*

$$I(R; X_d^{(c)} \mid X_s) = \frac{1}{2}\, \mathbb{E}\left[\log \frac{\sigma_0^2(X_s)}{\sigma_c^2(X_s, X_d^{(c)})}\right] = \frac{1}{2}\, \mathbb{E}\left[\log \frac{\mathrm{Var}(R \mid X_s)}{\mathrm{Var}(R \mid X_s, X_d^{(c)})}\right]. \tag{33}$$

**Proof.** For a Gaussian random variable $Z \sim \mathcal{N}(m, \sigma^2)$, $h(Z) = \frac{1}{2}\log(2\pi e \sigma^2)$. Applying this to $R \mid X_s$ and $R \mid (X_s, X_d^{(c)})$ and taking their difference yields the stated identity. $\square$

**Interpretation.** Under Gaussian posterior contraction, conditional mutual information is exactly the *expected log-shrinkage* of the posterior variance. In particular, positive $I(R; X_d^{(c)} \mid X_s)$ implies a multiplicative reduction in conditional variance on average, not merely an additive decrease.

More generally, without Gaussianity one can still obtain one-sided inequalities using the maximum entropy principle. Since for any random variable $Z$ with finite variance,

$$h(Z) \leq \frac{1}{2} \log \left(2\pi e \operatorname{Var}(Z)\right), \tag{34}$$

we obtain the following bound.

**Proposition B.5** (Entropy upper bound yields a conservative MI–variance relation). *Assume $R \mid X_s$ and $R \mid (X_s, X_d^{(c)})$ have finite conditional variances almost surely. Then*

$$I\big(R; X_d^{(c)} \mid X_s\big) \geq h(R \mid X_s) - \frac{1}{2} \, \mathbb{E}\Big[\log \Big(2\pi e \operatorname{Var}(R \mid X_s, X_d^{(c)})\Big)\Big]. \tag{35}$$

*In particular, any substantial decrease in $\operatorname{Var}(R \mid X_s, X_d^{(c)})$ forces a nontrivial lower bound on $I(R; X_d^{(c)} \mid X_s)$.*

**Proof.** By definition, $I(R; X_d^{(c)} \mid X_s) = h(R \mid X_s) - h(R \mid X_s, X_d^{(c)})$. Apply $h(\cdot) \leq \frac{1}{2} \log(2\pi e \operatorname{Var}(\cdot))$ to the second term and take expectation. $\square$

This proposition is deliberately conservative: it does not claim tightness, but it formalizes the direction that matters for our theory—variance reduction cannot occur without information gain.

### B.4. Information Revelation as Posterior Contraction Along Two Modes

We now formalize the "information revelation" viewpoint. Let

$$\pi_c(r) \triangleq p(R = r \mid \mathcal{I}_c) \tag{36}$$

denote the posterior distribution over outcomes given information at compute $c$. The Bayes risk under MSE is the expected posterior variance:

$$\mathcal{L}(c) = \mathbb{E}[\operatorname{Var}(R \mid \mathcal{I}_c)]. \tag{37}$$

Thus, decreasing $\mathcal{L}(c)$ corresponds to posterior contraction in the second moment sense.

*Remark* B.6 (Two modes of information revelation (formal view)). The filtration $\{\mathcal{I}_c\}_{c \geq 0}$ induces two distinct modes of posterior contraction:

1. **Ex-ante (static) contraction.** Conditioning on $X_s$ transitions the prior $p(R)$ to the static posterior $p(R \mid X_s)$, yielding an initial reduction in uncertainty from $\operatorname{Var}(R)$ to $\operatorname{Var}(R \mid X_s)$. This contraction occurs at $c = 0$ and reflects what can be inferred from the data–model pair *without* observing optimization.

2. **In-process (dynamic) contraction.** Conditioning further on $X_d^{(c)}$ transitions $p(R \mid X_s)$ to $p(R \mid X_s, X_d^{(c)})$, reducing the residual uncertainty attributable to stochastic optimization. This contraction unfolds continuously with $c$ and is quantified by $\mathcal{V}_{opt}(c)$ in Eq. (4).

In this view, pre-hoc prediction is not a discrete hyperparameter selection problem. It is a controlled posterior contraction process: the practitioner allocates computation to shrink $\operatorname{Var}(R \mid \mathcal{I}_c)$ until the marginal information gain becomes negligible relative to cost.

### B.5. A Useful Identity: Optimization Variance as an MMSE Gap

Finally, we record an identity that is often useful when connecting information gain to estimation performance. Recall from Appendix A that

$$\mathcal{L}(c) = \mathbb{E}[\operatorname{Var}(R \mid \mathcal{I}_c)]. \tag{38}$$

Define the (Bayes-optimal) minimum mean squared error (MMSE) at compute $c$ by

$$\mathrm{mmse}(c) \triangleq \mathbb{E}\big[(R - \mathbb{E}[R \mid \mathcal{I}_c])^2\big]. \tag{39}$$

Then $\mathrm{mmse}(c) = \mathcal{L}(c)$, and the optimization variance term can be written as

$$\mathcal{V}_{opt}(c) = \mathcal{L}(c) - \mathcal{L}_{int} = \mathrm{mmse}(c) - \mathrm{mmse}(\infty), \tag{40}$$

i.e., the *excess MMSE* incurred by observing only a finite trajectory prefix. This identity clarifies the functional role of dynamic probing: it reduces the MMSE gap left by static priors, and it does so only insofar as $I(R; X_d^{(c)} \mid X_s)$ is positive and grows with computation (Lemma B.3).

## C. Asymptotic Envelope for Uncertainty Decay

This appendix provides a concrete stochastic-approximation-based derivation illustrating the asymptotic uncertainty decay envelope discussed in Section 5. The purpose of this analysis is *not* to characterize the exact conditional variance given infinite information, but to demonstrate that under locally regular stochastic optimization dynamics, *observable uncertainty remaining at finite computation budgets cannot contract arbitrarily fast*. The assumptions adopted here are intentionally stronger than those in the main text and are used solely to make this rate-limiting mechanism explicit.

### C.1. Local Stochastic Approximation Regime

We consider an asymptotic regime of fine-tuning in which optimization proceeds within a locally stable neighborhood of a reference point $\theta^\star$. Let $\theta_t$ denote the parameter state after $t$ optimization updates, and let $c$ denote a continuous computation index proportional to $t$.

**Local regularity.** Assume that in a neighborhood of $\theta^\star$, the loss admits a second-order expansion

$$L(\theta) = L(\theta^\star) + \frac{1}{2}(\theta - \theta^\star)^\top H(\theta - \theta^\star) + o(\|\theta - \theta^\star\|^2), \tag{41}$$

where $H$ is symmetric and locally stable on the dynamically active subspace. No assumption is made about global convexity of the loss landscape.

**Stochastic optimization dynamics.** Consider the stochastic approximation recursion

$$\theta_{t+1} = \theta_t - \eta_t\big(\nabla L(\theta_t) + \xi_t\big), \tag{42}$$

where $\{\xi_t\}$ is a martingale difference sequence satisfying

$$\mathbb{E}[\xi_t \mid \mathcal{F}_t] = 0, \qquad \mathbb{E}[\xi_t \xi_t^\top \mid \mathcal{F}_t] = \Sigma_\xi, \tag{43}$$

with $\Sigma_\xi$ positive semidefinite and non-degenerate along at least one direction relevant to the evaluation metric.

**Step-size schedule.** Assume a polynomially decaying step size

$$\eta_t = \eta_0 t^{-\rho}, \qquad \rho \in \left(\tfrac{1}{2}, 1\right], \tag{44}$$

which ensures almost sure convergence while allowing persistent stochastic fluctuations over finite horizons.

### C.2. Finite-Horizon Parameter Uncertainty

Although $\theta_t$ converges almost surely to a random limit $\theta_\infty$, for any finite time $t$ there remains residual uncertainty about $\theta_\infty$ due to future stochastic gradient noise not yet revealed in $\mathcal{F}_t$. Unrolling the linearized error dynamics yields

$$\theta_\infty - \theta_t = -\sum_{k=t}^{\infty} \Phi_{\infty \leftarrow (k+1)}\, \eta_k\, \xi_k, \tag{45}$$

where $\Phi_{\infty \leftarrow (k+1)}$ denotes the state-transition operator of the linearized dynamics. Conditioned on $\mathcal{F}_t$, the future noise terms $\{\xi_k\}_{k \geq t}$ remain random, implying

$$\mathrm{Cov}(\theta_\infty \mid \mathcal{F}_t) \succeq \sum_{k=t}^{\infty} \eta_k^2 \, \Phi_{\infty \leftarrow (k+1)} \, \Sigma_\xi \, \Phi_{\infty \leftarrow (k+1)}^\top. \tag{46}$$

Local stability ensures that the transition operators preserve at least one noise-excited direction. Moreover, the tail of the squared step sizes satisfies

$$\sum_{k=t}^{\infty} \eta_k^2 = \Theta\left(t^{-(2\rho-1)}\right), \tag{47}$$

which yields a constant $K_\theta > 0$ such that

$$\mathbb{E}\left[\mathrm{Tr}\left(\mathrm{Cov}(\theta_\infty \mid \mathcal{F}_t)\right)\right] \geq K_\theta \, t^{-(2\rho-1)}. \tag{48}$$

### C.3. From Parameter Fluctuations to Observable Outcome Variability

Let the evaluation outcome be modeled as

$$R = R(\theta_\infty, \zeta), \tag{49}$$

where $\zeta$ represents task-level or evaluation-level randomness independent of the optimization noise. Assume that $R(\theta, \zeta)$ is locally smooth in $\theta$ and that $\nabla_\theta R(\theta^\star, \zeta)$ has non-zero projection onto a noise-excited direction with non-zero probability.

Conditioned on $\mathcal{F}_t$, uncertainty in $\theta_\infty$ induces variability in the conditional distribution of $R$. A first-order expansion yields

$$\mathrm{Var}(R \mid \mathcal{F}_t) \gtrsim \nabla R(\theta^\star)^\top \mathrm{Cov}(\theta_\infty \mid \mathcal{F}_t) \nabla R(\theta^\star) + \mathrm{Var}(\zeta), \tag{50}$$

where the second term represents irreducible task-level variability. Taking expectations and combining with (48) yields

$$\mathbb{E}[\mathrm{Var}(R \mid \mathcal{F}_t)] \gtrsim K_R \, t^{-(2\rho-1)} + \text{(irreducible terms)}. \tag{51}$$

### C.4. Implication for the Optimization-Induced Uncertainty Envelope

Recall that the optimization-induced uncertainty $\mathcal{V}_{opt}(c)$ captures the excess uncertainty attributable to observing only a finite prefix of the optimization trajectory. Identifying $\mathcal{I}_c$ with $\mathcal{F}_t$ up to reparameterization, the residual uncertainty arising from future stochastic fluctuations yields an asymptotic envelope

$$\mathcal{V}_{opt}(c) \gtrsim K \, c^{-\alpha}, \qquad \alpha = 2\rho - 1 \in (0, 1], \tag{52}$$

for some constant $K > 0$. We emphasize that this expression characterizes a *rate-limiting asymptotic behavior* of observable uncertainty contraction, rather than a tight bound on the conditional variance given infinite information.

The explicit form $\alpha = 2\rho - 1$ arises from the specific stochastic approximation regime analyzed here. In practice, different optimization schedules, preconditioning, and non-stationary phases can induce substantially different effective decay rates, consistent with the empirical observations reported in Section 8.

## D. Proofs for Budget-Optimal Allocation and the Efficiency Frontier

This appendix provides the detailed derivations and proofs for the budget-optimal allocation results in Section 6. We emphasize that the core object being optimized is the *reducible uncertainty* $\mathcal{V}_{opt}(c)$ governed by the uncertainty decay dynamics in Section 5, while the intrinsic limit $\mathcal{L}_{int}$ remains invariant to compute. We present results in increasing generality: first for a general monotone cost model, then for the linear cost specialization used in the main text, and finally for the geometry of the efficiency frontier.

## D.1. General Budgeted Risk Minimization

Recall the Bayes risk decomposition (Proposition 4.1):

$$\mathcal{L}(c) = \mathcal{L}_{int} + \mathcal{V}_{opt}(c), \tag{53}$$

where $\mathcal{L}_{int}$ is constant in $c$ and $\mathcal{V}_{opt}(c)$ is non-increasing in $c$ (Appendix A). Let $C(c)$ denote the cost of acquiring $\mathcal{I}_c$ and assume:

**Assumption D.1** (Admissible cost model). The cost function $C : [0, \infty) \to [0, \infty)$ is continuous, strictly increasing, and satisfies $C(0) = C_s$.

Given a total budget $B \geq C_s$, the decision problem is

$$\min_{c \geq 0} \ \mathcal{L}_{int} + \mathcal{V}_{opt}(c) \quad \text{s.t.} \quad C(c) \leq B. \tag{54}$$

**Lemma D.2** (Budget is optimally saturated). *Assume $\mathcal{V}_{opt}(c)$ is non-increasing and $C(c)$ is strictly increasing. Then any optimizer $c^\star$ of (54) satisfies*

$$C(c^\star) = B, \tag{55}$$

*unless $\mathcal{V}_{opt}(c)$ is flat on $[0, \bar{c}]$ for some $\bar{c} > 0$ and $B$ is large enough to exceed $C(\bar{c})$.*

**Proof.** If $C(c^\star) < B$, then there exists $\delta > 0$ such that $C(c^\star + \delta) \leq B$ by continuity and strict monotonicity of $C(\cdot)$. Since $\mathcal{V}_{opt}(\cdot)$ is non-increasing, $\mathcal{V}_{opt}(c^\star + \delta) \leq \mathcal{V}_{opt}(c^\star)$, yielding a weakly better feasible point. Strict improvement holds unless $\mathcal{V}_{opt}$ is locally flat. $\square$

Lemma D.2 formalizes the economic intuition: if probing is beneficial (strictly reduces uncertainty), one should use the full available budget.

## D.2. KKT Derivation Under a Canonical Power-Law Model

The main text adopts a canonical model for the reducible uncertainty:

$$\mathcal{V}_{opt}(c) = Kc^{-\alpha}, \qquad K > 0, \ \alpha > 0, \tag{56}$$

valid in an early-to-mid regime beyond a minimal onset $c_0$ (Section 5). We first derive the optimal allocation under a general differentiable cost model.

**Assumption D.3** (Differentiable cost). $C(\cdot)$ is differentiable and strictly increasing, with derivative $C'(c) > 0$ for all $c \geq 0$.

Consider the Lagrangian for (54) under (56):

$$\mathcal{J}(c, \gamma) = \mathcal{L}_{int} + Kc^{-\alpha} + \gamma \left( C(c) - B \right), \qquad \gamma \geq 0. \tag{57}$$

**Theorem D.4** (KKT characterization for budget-optimal probing). *Under Assumptions D.1 and D.3, and $\mathcal{V}_{opt}(c) = Kc^{-\alpha}$ with $\alpha > 0$, any interior optimum $c^\star > 0$ satisfies the KKT stationarity condition*

$$\left| \frac{d}{dc} \mathcal{V}_{opt}(c) \right|_{c=c^\star} = \gamma^\star C'(c^\star), \tag{58}$$

*together with complementary slackness*

$$\gamma^\star \left( C(c^\star) - B \right) = 0, \qquad \gamma^\star \geq 0, \qquad C(c^\star) \leq B. \tag{59}$$

**Proof.** This is a standard application of KKT conditions for a single inequality constraint. Since $C(\cdot)$ is differentiable and strictly increasing, the constraint qualification holds. Stationarity requires $\partial \mathcal{J}/\partial c = 0$:

$$-\alpha Kc^{-(\alpha+1)} + \gamma C'(c) = 0,$$

equivalently $\alpha Kc^{-(\alpha+1)} = \gamma C'(c)$, which is (58) since $\mathcal{V}_{opt}$ is decreasing. The remaining KKT conditions are feasibility and complementary slackness. $\square$

**Economic interpretation.** Equation (58) states a marginal equilibrium: the *rate of uncertainty reduction* per unit compute equals the *shadow-price-adjusted marginal cost*. This is the precise sense in which optimal probing is not a fixed step count: it is the equilibrium point induced by the governing dynamics (through $\alpha$ and $K$) and the cost geometry (through $C'(c)$).

### D.3. Closed-Form Solution Under Linear Cost

We now specialize to the linear cost model used in Section 6:

$$C(c) = C_s + \lambda c, \qquad \lambda > 0. \tag{60}$$

The feasible set is $0 \le c \le (B - C_s)/\lambda$.

**Corollary D.5** (Budget-optimal compute under linear cost). *Under* (56) *and* (60)*, the optimizer is*

$$c^\star = \frac{B - C_s}{\lambda}, \tag{61}$$

*and the achieved risk is*

$$\mathcal{L}(c^\star) = \mathcal{L}_{int} + K \left( \frac{B - C_s}{\lambda} \right)^{-\alpha}. \tag{62}$$

**Proof.** Since $\mathcal{V}_{opt}(c) = Kc^{-\alpha}$ is strictly decreasing in $c$ for $\alpha > 0$ and the feasible set is an interval, the minimizer is the largest feasible $c$. Under (60) this is exactly (61). Substituting into $\mathcal{L}(c)$ yields (62). $\square$

**Remark (why the main text may introduce a stopping rule).** Corollary D.5 formalizes the simplest budgeted setting: if the only objective is to reduce prediction risk and the cost is linear, then one uses the full budget (Lemma D.2). In practice, one often introduces an additional decision criterion, such as minimizing a *joint* objective that trades risk against compute (Appendix D.4), or stopping when marginal gains fall below a tolerance (Appendix C). The "optimal stopping" viewpoint in the main text corresponds to these economically meaningful variants.

### D.4. Risk–Cost Tradeoff Formulation and the Equilibrium Condition

A complementary way to formalize "optimal stopping" is to treat compute as costly directly in the objective. Define the tradeoff objective

$$\min_{c \ge 0} \ \mathcal{L}(c) + \gamma \, C(c), \qquad \gamma > 0, \tag{63}$$

where $\gamma$ controls the exchange rate between risk and compute. This formulation is equivalent to the constrained problem (54) via Lagrangian duality and yields a clean equilibrium condition.

**Proposition D.6** (Equilibrium condition under risk–cost tradeoff). *Under* (56) *and differentiable* $C(\cdot)$ *with* $C'(c) > 0$*, any interior minimizer* $c^\star > 0$ *of* (63) *satisfies*

$$\left| \frac{d}{dc} \mathcal{V}_{opt}(c) \right|_{c=c^\star} = \gamma \, C'(c^\star). \tag{64}$$

*In particular, under linear cost* $C'(c) = \lambda$*,*

$$c^\star = \left( \frac{\alpha K}{\gamma \lambda} \right)^{\frac{1}{\alpha+1}}. \tag{65}$$

**Proof.** The derivative of the objective in (63) is

$$\frac{d}{dc} \mathcal{L}(c) + \gamma C'(c) = \frac{d}{dc} \mathcal{V}_{opt}(c) + \gamma C'(c).$$

Setting this to zero yields the equilibrium condition, and substituting $\mathcal{V}_{opt}(c) = Kc^{-\alpha}$ gives $-\alpha Kc^{-(\alpha+1)} + \gamma C'(c) = 0$. For linear cost, solving yields (65). $\square$

**Connection to Theorem D.4.** Proposition D.6 can be viewed as the unconstrained primal associated with the KKT multiplier interpretation: $\gamma$ plays the role of a shadow price for compute, and (65) matches the stationarity form used in the main text.

## D.5. Efficiency Frontier Geometry

We now characterize the Pareto frontier in the risk–cost plane. Let $c$ index operating points. Under (60), define

$$x(c) \triangleq C(c) = C_s + \lambda c, \qquad y(c) \triangleq \mathcal{L}(c) = \mathcal{L}_{int} + Kc^{-\alpha}. \tag{66}$$

Eliminating $c = (x - C_s)/\lambda$ yields an explicit frontier:

$$y(x) = \mathcal{L}_{int} + K \left( \frac{x - C_s}{\lambda} \right)^{-\alpha}, \qquad x > C_s. \tag{67}$$

**Proposition D.7** (Frontier slope and curvature). *For $x > C_s$, the frontier (67) is strictly decreasing and convex. Its slope and curvature are*

$$\frac{dy}{dx} = -\alpha K \lambda^\alpha (x - C_s)^{-(\alpha+1)}, \tag{68}$$

$$\frac{d^2y}{dx^2} = \alpha(\alpha + 1) K \lambda^\alpha (x - C_s)^{-(\alpha+2)}. \tag{69}$$

**Proof.** Differentiate (67) directly:

$$y(x) = \mathcal{L}_{int} + K\lambda^\alpha (x - C_s)^{-\alpha}.$$

Then

$$\frac{dy}{dx} = -\alpha K\lambda^\alpha (x - C_s)^{-(\alpha+1)} < 0,$$

and

$$\frac{d^2y}{dx^2} = \alpha(\alpha + 1)K\lambda^\alpha (x - C_s)^{-(\alpha+2)} > 0.$$

Thus the frontier is decreasing and convex. $\square$

**Interpretation.** Convexity formalizes diminishing returns: as cost increases, each additional unit of compute yields smaller risk reduction. The exponent $\alpha$ governs the global shape: smaller $\alpha$ produces a "long tail" frontier where uncertainty resolves slowly, while larger $\alpha$ yields a sharper early drop.

## D.6. Minimal Effective Regime via a Marginal-Gain Criterion

A step-count-free definition of an "effective probing regime" can be derived by thresholding the marginal gain. Under linear cost, the marginal risk reduction per unit cost equals

$$\left| \frac{dy}{dx} \right| = \alpha K \lambda^\alpha (x - C_s)^{-(\alpha+1)}. \tag{70}$$

Given a tolerance $\varepsilon > 0$ (risk reduction per unit cost), define the effective boundary $x_{\text{eff}}(\varepsilon)$ as the smallest cost beyond which the marginal gain falls below $\varepsilon$:

$$x_{\text{eff}}(\varepsilon) \triangleq \inf \left\{ x > C_s : \left| \frac{dy}{dx} \right| \leq \varepsilon \right\}. \tag{71}$$

Solving yields

$$x_{\text{eff}}(\varepsilon) = C_s + \left( \frac{\alpha K \lambda^\alpha}{\varepsilon} \right)^{\frac{1}{\alpha+1}}. \tag{72}$$

Equivalently, in compute units,

$$c_{\text{eff}}(\varepsilon) = \frac{x_{\text{eff}}(\varepsilon) - C_s}{\lambda} = \left( \frac{\alpha K}{\varepsilon \lambda} \right)^{\frac{1}{\alpha+1}}, \tag{73}$$

which matches the structural "elbow" threshold derived from the governing dynamics (cf. Appendix C). This criterion depends on $\alpha$ and $K$ rather than any fixed step count.

**D.7. Optional Extension: Joint Allocation Between Static and Dynamic Acquisition**

The main text emphasizes that static and dynamic information play distinct functional roles (Section 4). For completeness, we record a simple joint allocation model in which the practitioner can invest in static acquisition (e.g., richer static descriptors) at an explicit cost.

Let $m \geq 0$ denote a scalar control for static acquisition (e.g., feature complexity, evaluation resolution), with cost $C_s(m)$ and intrinsic limit $\mathcal{L}_{int}(m)$ decreasing in $m$. The joint allocation problem is

$$\min_{m \geq 0, \, c \geq 0} \mathcal{L}_{int}(m) + \mathcal{V}_{opt}(c) \quad \text{s.t.} \quad C_s(m) + C_d(c) \leq B. \tag{74}$$

Assuming differentiability and an interior solution, the KKT conditions yield the marginal-equalization principle:

$$-\frac{d}{dm}\mathcal{L}_{int}(m^\star) = \gamma^\star C'_s(m^\star), \qquad \left|\frac{d}{dc}\mathcal{V}_{opt}(c)\right|_{c=c^\star} = \gamma^\star C'_d(c^\star), \tag{75}$$

with a shared multiplier $\gamma^\star$. Equation (75) states that, at optimum, each dollar of budget buys the same reduction in risk regardless of whether it is spent on improving static priors or extending dynamic observation. This recovers the qualitative guidance emphasized in the main text: budgets should be allocated according to *marginal* risk reduction, and the governing dynamics parameter $\alpha$ directly controls the dynamic marginal returns.

# E. Identifiability of Governing Dynamics and Regime Structure

This appendix addresses the *identifiability* of the governing quantities introduced in the main text, namely the effective uncertainty decay rate $\alpha$ and the intrinsic risk floor $\mathcal{L}_{\mathcal{T},int}$. Our goal is to establish that these quantities are *empirically characterizable as structural properties* of a task–optimizer configuration, using finite observations collected offline.

Importantly, this appendix does **not** propose a deployment-time estimation protocol for individual fine-tuning runs. Rather, $\alpha$ and $\mathcal{L}_{\mathcal{T},int}$ are treated as task-level descriptors, analogous to learning-curve exponents in neural scaling laws, which are typically estimated under controlled experimental conditions and reused for analysis and comparison.

## E.1. Identifiability of the Uncertainty Decay Rate

As shown in Section 5, optimization-induced uncertainty obeys a rate-limiting asymptotic envelope of the form

$$\mathcal{V}_{opt}(c) \gtrsim K c^{-\alpha}. \tag{76}$$

This result characterizes how rapidly uncertainty can contract with increasing computation, but does not require direct observation of $\mathcal{V}_{opt}(c)$ itself.

**Observable uncertainty proxies.** In practice, we observe surrogate quantities that reflect optimization-induced variability under controlled randomness. For a fixed task–optimizer configuration $\mathcal{T} = (M, D, \mathcal{A})$ and a probing budget $c$, consider repeated probes that differ only in stochastic optimization noise (e.g., random seeds). Let $U_i(c)$ denote an observable outcome proxy derived from the $i$-th probe, such as prediction dispersion or run-to-run variability. An empirical uncertainty proxy is then given by

$$\widehat{U}(c) = \frac{1}{N-1} \sum_{i=1}^{N} \left(U_i(c) - \overline{U}(c)\right)^2, \qquad \overline{U}(c) = \frac{1}{N} \sum_{i=1}^{N} U_i(c). \tag{77}$$

While $\widehat{U}(c)$ does not equal $\mathcal{V}_{opt}(c)$, it is required only to preserve the relative decay behavior induced by optimization dynamics.

**Scaling-based identification.** Once the optimization enters a regime in which uncertainty contraction is rate-limited, the effective decay rate $\alpha$ can be identified by fitting the surrogate proxy to a power-law envelope:

$$\log \widehat{U}(c) = \log K - \alpha \log c + \varepsilon(c), \tag{78}$$

where $\varepsilon(c)$ captures finite-sample effects and higher-order deviations. This establishes identifiability of $\alpha$ as a *scaling exponent of observable uncertainty*, without implying that such fitting must be performed online or per deployment instance.

**Onset of the scaling regime.** For small values of $c$, transient optimization dynamics may violate the asymptotic assumptions underlying the envelope. Accordingly, identification of $\alpha$ is assessed over a computation range $[c_{\min}, c_{\max}]$ in which empirical curvature stabilizes. This choice affects estimation variance but does not alter the qualitative regime structure defined below.

### E.2. Identifiability of the Intrinsic Risk Floor

The intrinsic limit $\mathcal{L}_{\mathcal{T},int}$ represents the irreducible component of task-level prediction risk, corresponding to uncertainty that persists even under asymptotically complete information. In empirical analysis, $\mathcal{L}_{\mathcal{T},int}$ is not observed directly, but can be inferred as the intercept of the fitted uncertainty envelope.

Specifically, given a set of uncertainty proxies $\{\widehat{U}(c_i)\}$, we jointly fit the model

$$\widehat{U}(c) \approx \mathcal{L}_{\mathcal{T},int} + Kc^{-\alpha}, \tag{79}$$

thereby avoiding reliance on saturation at a finite probing depth. The resulting $\widehat{\mathcal{L}}_{\mathcal{T},int}$ should be interpreted as an *effective intrinsic floor* consistent with the observable uncertainty decay, rather than as a precise asymptotic limit.

### E.3. Regime Structure and Assignment

Given identifiable estimates $(\widehat{\alpha}, \widehat{\mathcal{L}}_{\mathcal{T},int})$, tasks can be positioned within the predictability phase space introduced in Section 8.3. The regimes defined in the main text correspond to qualitative regions of this space, rather than sharp decision boundaries or deployment-time classifiers.

**Static-Sufficient Regime.** Tasks with rapid uncertainty contraction (large $\widehat{\alpha}$) or a dominant intrinsic floor exhibit minimal benefit from extended probing. In this regime, static information suffices for effective prediction.

**Dynamic-Critical Regime.** Tasks characterized by slower uncertainty decay and a low intrinsic floor possess substantial reducible uncertainty. Here, dynamic observation plays a decisive role in resolving outcome variability, and the efficiency frontier extends deep into the computational budget.

**Noise-Dominant Regime.** Tasks with a large intrinsic floor remain dominated by irreducible ambiguity, such that additional probing yields diminishing returns regardless of depth.

### E.4. Interpretation and Scope

We emphasize that regime assignment is inherently comparative and descriptive. It does not require precise numerical thresholds, nor does it assume that governing parameters must be estimated online for each deployment instance. Instead, the phase diagram provides a *mechanism-based organization* of predictability behavior across tasks, grounded in empirically identifiable uncertainty dynamics.

This distinction separates the theoretical limits analyzed in this work from questions of system design or real-time decision-making, which lie beyond the present scope.

## F. Additional Experimental Details

This appendix provides supplementary experimental details and analyses omitted from the main text due to space constraints. Our goal is to ensure reproducibility of the empirical results in Section 8 and to clarify how the theoretical quantities introduced in the framework are instantiated through *observable uncertainty proxies*.

We do not assume that the maximal probing depth $c_{\max}$ reaches the true asymptotic limit $c \to \infty$. All reported quantities should be interpreted as effective parameters inferred from the stable decay regime observable within the available computational budget.

All analyzes in this appendix support, but do not extend, the claims made in the main text.

### F.1. Tasks and Regime Coverage

We evaluate pre-hoc fine-tuning predictability across a diverse collection of downstream tasks and pretrained foundation models, spanning different objectives, data conditions, model scales, and optimization behaviors. Tasks are selected to ensure coverage of all three predictability regimes introduced in Section 8.3.

**Foundation models.** Our experiments include multiple instruction-tuned large language models across different model families and parameter scales. Specifically, we evaluate models from the *Qwen* family (*Qwen-0.5B*, *Qwen-2.5B*, and *Qwen-7B-Instruct (Qwen et al., 2025)*), as well as *LLaMA-3-8B-Instruct (Team, 2024)* and *Mistral-7B (Jiang et al., 2023)*. This selection enables analysis across both scaling dimensions and heterogeneous pretraining and instruction-tuning pipelines.

**Downstream tasks.** The evaluated tasks span classification, reasoning, and generation objectives. They include sentiment classification (*SST-2 (Wang et al., 2019)*), arithmetic and multi-step reasoning (*GSM8K (Cobbe et al., 2021)*), broad multi-domain knowledge evaluation (*MMLU (Hendrycks et al., 2021)*), abstractive summarization (*SAMSum (Gliwa et al., 2019)*), and robustness-oriented truthfulness evaluation (*TruthfulQA (Lin et al., 2022)*).

The benchmark suite is intended to be representative rather than exhaustive. Complementary stress tests for pre-hoc predictability include statistical reasoning, text-to-SQL and semantic-error detection, cross-document multi-entity question answering, retrieval-augmented generation, and chart question answering (Zhu et al., 2024; Liu et al., 2025; Lin et al., 2025a; Yang et al., 2024; Wu et al., 2024). These settings are promising candidates for future analysis because they expose heterogeneous sources of uncertainty, including reasoning difficulty, retrieval incompleteness, entity-level attribution errors, and semantic mismatch.

**Optimization settings.** Unless otherwise stated, fine-tuning is performed using AdamW optimizer (Loshchilov & Hutter, 2019) with task-specific learning-rate schedules and batch sizes. We employ LoRA (Hu et al., 2021) for parameter-efficient fine-tuning (rank=32, alpha=64, dropout=0.1) across all experiments to ensure computational feasibility at scale. All hyperparameters are held fixed within each task-model configuration to isolate variability induced by stochastic optimization. Experiments are conducted in bf16 precision using the Hugging Face Transformers library (Wolf et al., 2020).

Regime assignments are determined *post hoc* based on the estimated governing descriptors $(\widehat{\alpha}, \widehat{\mathcal{L}}_{\mathcal{T},int})$. They are used exclusively for analysis and visualization and are not employed to tune probing strategies, predictors, or hyperparameters.

### F.2. Probing Protocol and Randomization

For each task, probing is performed by executing partial fine-tuning runs up to a computation budget $c$, measured in optimization steps. The probing budget set $\mathcal{C}$ spans early, intermediate, and late stages of fine-tuning.

To isolate optimization-induced variability, we perform repeated probing runs with identical static configurations while varying stochastic sources such as random seeds, data ordering, and stochastic framework operations. Unless otherwise stated, all other hyperparameters are held fixed within each task.

Each task-budget pair is evaluated with $N = 1{,}500$ independent runs. This large sample size is used solely to support *offline identifiability analysis* and does not reflect requirements for deployment-time usage.

### F.3. Observable Uncertainty Proxies

At each probing depth $c$, we compute an observable uncertainty proxy

$$\widehat{U}(c) = \frac{1}{N-1} \sum_{i=1}^{N} \left( R_i(c) - \overline{R}(c) \right)^2, \qquad \overline{R}(c) = \frac{1}{N} \sum_{i=1}^{N} R_i(c), \tag{80}$$

where $R_i(c)$ denotes the realized final fine-tuning outcome of the $i$-th probing run. This proxy reflects dispersion induced by stochastic optimization and serves as a measurable surrogate for uncertainty relevant to prediction.

We emphasize that $\widehat{U}(c)$ is *not* assumed to equal the theoretical quantity $\mathcal{V}_{opt}(c)$, but is required only to preserve relative decay behavior across $c$. To mitigate finite-sample noise, we enforce monotonicity of $\widehat{U}(c)$ with respect to $c$ using isotonic regression, consistent with the information filtration assumption.

## F.4. Predictors Used for Empirical Evaluation

All empirical evaluations involve a predictor $f$ that maps the available information at probing depth $c$ to an estimate of the final fine-tuning outcome. Across all experiments, the predictor architecture and training procedure are held fixed; only the information sources provided as input vary.

We consider three predictor families:

- **Static-only**: the predictor uses only static descriptors available prior to fine-tuning (e.g., dataset statistics and model–data compatibility features).

- **Dynamic-only**: the predictor uses only dynamic signals extracted from partial optimization trajectories up to depth $c$ (e.g., early loss curves and gradient statistics).

- **Hybrid**: the predictor combines both static and dynamic features.

The predictor is trained to minimize squared error against the final outcome across repeated runs. Differences in empirical performance therefore reflect information availability rather than model capacity.

## F.5. Estimating Governing Parameters

The governing parameters $(\alpha, \mathcal{L}_{\mathcal{T}, int})$ are estimated by jointly fitting the uncertainty envelope

$$\widehat{U}(c) \approx \mathcal{L}_{\mathcal{T}, int} + K c^{-\alpha} \tag{81}$$

over a stable probing interval $[c_{\min}, c_{\max}]$. This joint fitting procedure avoids reliance on saturation at a finite probing depth and aligns with the identifiability analysis in Appendix E.

The lower bound $c_{\min}$ excludes transient early dynamics, while $c_{\max}$ corresponds to the largest probing depth considered.

In practice, the lower bound $c_{\min}$ is selected as the smallest probing depth after which the log–log plot of $\widehat{U}(c)$ exhibits approximately linear behavior. Operationally, this corresponds to excluding the first one or two probing points.

We verify that estimated decay rates and regime assignments are robust to moderate variations of this choice.

## F.6. Normalization and Aggregation Across Tasks

To aggregate results across tasks with different absolute scales, all uncertainty-related quantities are normalized on a per-task basis. Specifically, observable uncertainty proxies $\widehat{U}(c)$ are normalized by their value at the smallest probing depth $c_0$ for each task:

$$\widehat{U}_{\mathrm{norm}}(c) := \frac{\widehat{U}(c)}{\widehat{U}(c_0)}.$$

The marginal gain shown in aggregated figures is defined as

$$\Delta(c) := \widehat{U}_{\mathrm{norm}}(c) - \widehat{U}_{\mathrm{norm}}(c + \Delta c),$$

which measures the reduction in normalized uncertainty achieved by additional probing. This normalization ensures comparability across tasks while preserving relative decay behavior.

## F.7. Probe-Budget Sensitivity

To complement the efficiency-frontier analysis in Section 8.4, we further examine how regime-level decisions evolve as the probing depth increases. This analysis is intended to characterize the stability of coarse regime assignments under different probe budgets, rather than to identify a universally optimal number of probing steps.

Table 1 reports three quantities across increasing probe depths. First, regime agreement with the full-fit reference measures the fraction of tasks whose regime assignment matches the reference assignment obtained from the full offline fitting protocol. Second, the observable uncertainty proxy measures the remaining normalized uncertainty estimated at each

*Table 1.* Probe-budget sensitivity analysis. Regime-level decisions stabilize under moderate probing depth, while deeper probing mainly refines uncertainty estimates at higher cost.

| Metric | 25 | 50 | 75 | 100 | 150 | 200 |
|---|---|---|---|---|---|---|
| Regime agreement vs. full-fit reference (%) | 69 | 78 | 85 | 90 | 91 | 90 |
| Observable uncertainty proxy (%) | 29.8 | 22.1 | 18.7 | 11.5 | 10.2 | 9.7 |
| Relative probe cost (100-step = 1.0) | 0.16 | 0.52 | 0.86 | 1.00 | 1.34 | 1.71 |

probing depth, where lower values indicate more refined uncertainty estimation. Third, relative probe cost reports the wall-clock cost normalized so that the 100-step probe has cost $1.0$.

The results show a clear stabilization pattern. Regime agreement increases rapidly from $69\%$ at 25 steps to $90\%$ at 100 steps, after which deeper probing yields only marginal changes. In contrast, relative probe cost continues to increase from $1.00$ at 100 steps to $1.34$ at 150 steps and $1.71$ at 200 steps. The observable uncertainty proxy also continues to decrease beyond 100 steps, but at a slower rate. This suggests that deeper probing can still refine uncertainty estimates, while its marginal benefit for regime-level decisions becomes limited once the stable probing interval has been reached.

These results provide an operational interpretation of lightweight probing. A probe is lightweight not because it uses a fixed universal number of steps, but because it is short relative to full fine-tuning while being long enough to recover the regime-relevant structure of uncertainty decay. In our experiments, moderate probe depths around 100 steps already recover stable regime-level behavior, whereas substantially deeper probes mainly improve the precision of the surrogate uncertainty estimate.

### F.8. Supplementary Evidence for Intrinsic Risk Floors

Figure 5 plots the observable uncertainty proxy $\widehat{U}(c)$ as a function of probing depth, aggregated by regime. Noise-Dominant regimes exhibit substantially higher plateaus, while Static-Sufficient and Dynamic-Critical regimes converge toward lower effective intrinsic floors. These trends provide empirical consistency evidence for the intrinsic-risk interpretation underlying the phase diagram.

Such effective floors can arise from several data-centric sources studied in prior work, including mislabels, missing or incomplete values, limited redundancy, data scarcity, and broader data-quality pathologies (Yang et al., 2025; Chai et al., 2024; Fan et al., 2013). These sources do not necessarily vanish with additional optimization steps, and therefore align naturally with the intrinsic-limited interpretation of the Noise-Dominant regime.

### F.9. Population-Level Distribution of Decay Rates

Figure 6 shows the distribution of estimated decay exponents $\widehat{\alpha}$ across tasks, grouped by regime. Dynamic-Critical tasks concentrate on lower $\widehat{\alpha}$ values, while Static-Sufficient tasks exhibit systematically higher decay rates. This separation supports the interpretation of $\alpha$ as a descriptor of dynamic difficulty rather than as an artifact of a small number of outliers.

### F.10. Ablation of Information Sources

Figure 7 compares predictive performance under a fixed probing budget for three estimator families: static-only, dynamic-only, and hybrid. Dynamic information is beneficial primarily in Dynamic-Critical regimes, while Static-Sufficient and Noise-Dominant regimes show limited gains, consistent with the regime semantics.

### F.11. Hard-Case Localization

Figure 8 overlays empirically hard-to-predict cases on the phase diagram defined by $(\widehat{\mathcal{L}}_{\mathcal{T},int}, \widehat{\alpha})$. Hard cases cluster in Noise-Dominant regimes or near regime boundaries, supporting the interpretation that empirical failures arise from regime mismatch rather than estimator inadequacy.

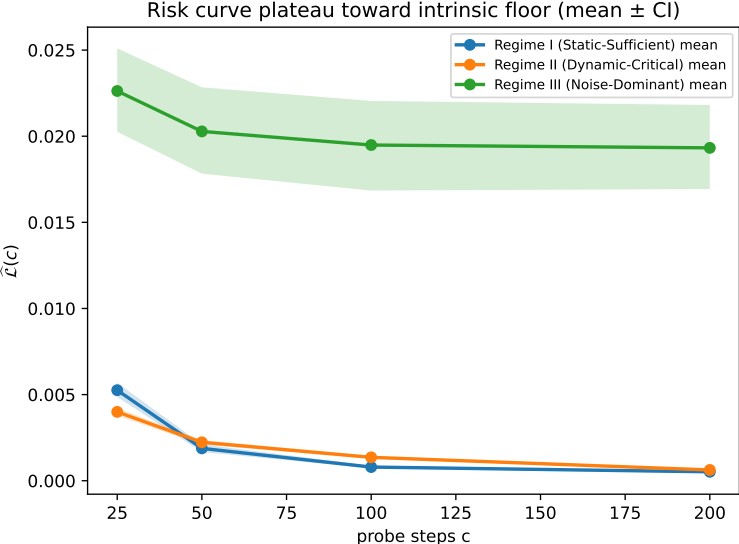

*Figure 5.* Observable uncertainty decay and effective intrinsic floors across regimes. Mean $\widehat{U}(c)$ with confidence intervals is shown as a function of probing depth. Noise-Dominant regimes plateau at higher levels, indicating larger intrinsic ambiguity.

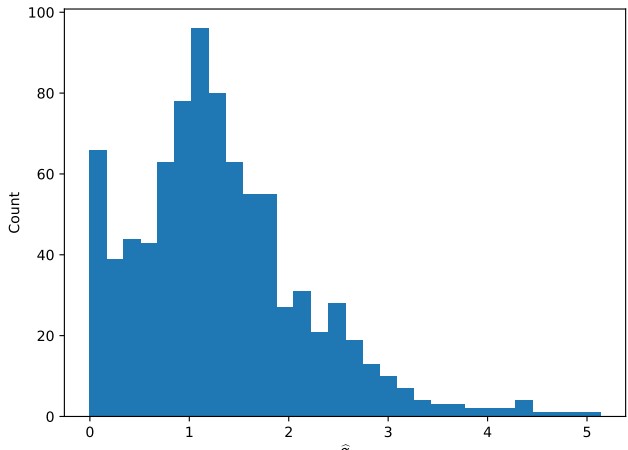

*Figure 6.* Distribution of estimated uncertainty decay rates across regimes. Dynamic-Critical tasks exhibit slower decay, while Static-Sufficient tasks show rapid contraction.

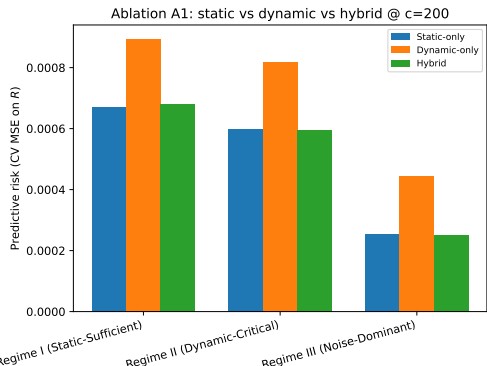

*Figure 7.* Ablation of information sources under a fixed probing budget. Dynamic signals are informative only in Dynamic-Critical regimes.

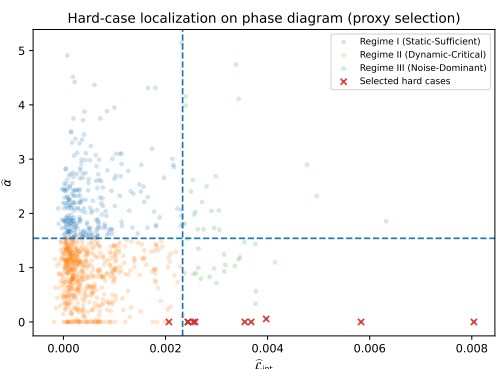

*Figure 8.* Localization of hard cases on the empirical phase diagram. Hard-to-predict tasks cluster in Noise-Dominant regimes or near regime boundaries.

