# OpenReview forum: "A Risk Decomposition Framework for Pre-hoc Fine-tuning Prediction"
_ICML.cc/2026/Conference — ICML 2026 regular_

### Official Review · Reviewer_P983 · 2026-03-11

**Soundness:** 3
**Presentation:** 3
**Significance:** 2
**Originality:** 4
**Overall Recommendation:** 4
**Confidence:** 2

**Summary:**

Pre-hoc prediction of fine-tuning outcomes in large language model adaptation is examined through a theoretical framework that decomposes prediction risk into two components: an intrinsic limit determined by static task properties and an optimization-induced variance that decreases with probing computation. Pre-hoc prediction is formulated as a stochastic estimation problem under progressively revealed information, leading to a risk decomposition derived from the law of total variance and an analysis of uncertainty reduction dynamics during probing. Based on this formulation, a budget-aware probing principle is derived to characterize the trade-off between prediction risk and computational cost. Empirical evaluation across multiple pretrained LLMs and downstream tasks analyzes uncertainty decay behavior and reveals distinct predictability regimes consistent with the theoretical analysis.

**Compliance With Llm Reviewing Policy:**

Affirmed.

**Key Questions For Authors:**

1. The framework relies on estimating quantities such as the effective uncertainty decay rate and the intrinsic risk floor. In the experiments, these quantities are obtained using repeated runs and offline fitting procedures. How can these parameters be estimated reliably in practical scenarios where only a limited number of probing runs are available?
2. How would the proposed framework integrate with existing proxy-based or early-trajectory prediction methods? For instance, could the risk decomposition perspective be used to guide the design or improvement of practical predictors used in real training pipelines?
3. The study derives a budget-aware probing condition based on the trade-off between marginal uncertainty reduction and computational cost. Have the authors evaluated whether this stopping rule leads to meaningful compute savings or improved decision-making in realistic training settings?

**Limitations:**

yes

**Strengths And Weaknesses:**

Strengths:
1. The work reframes pre-hoc fine-tuning prediction as a risk decomposition problem rather than a purely empirical regression task. The separation of prediction error into an intrinsic limit and an optimization variance provides a clear conceptual lens for understanding the fundamental limits of predictability.
2. The study presents a structured theoretical analysis of the problem. In particular, it derives a risk decomposition based on conditional variance, establishes an asymptotic envelope governing the decay of optimization-induced uncertainty, and formulates probing as a risk–cost trade-off that yields a budget-aware probing condition.

Weaknesses:
1. Several theoretical results rely on simplifying assumptions about the optimization dynamics. While these assumptions enable analytical tractability, it is unclear how well they hold in realistic large-scale fine-tuning settings, especially during early training phases or under highly non-convex optimization landscapes.
2. The empirical evaluation mainly focuses on validating structural properties predicted by the theory (e.g., uncertainty decay patterns and regime structure) rather than comparing against existing state-of-the-art pre-hoc prediction methods. Consequently, the practical advantage of the proposed framework relative to prior approaches remains difficult to assess.
3. Important parameters such as the intrinsic risk floor and the uncertainty decay rate must be estimated through offline fitting procedures. The feasibility and robustness of estimating these quantities reliably in realistic deployment settings are not demonstrated.

---

> ### Author Rebuttal · Authors · 2026-03-27
>
> We thank the reviewer for the careful reading, the detailed summary, and especially for recognizing the paper's originality. We also appreciate these very helpful suggestions and we will revise the final version accordingly.
>
> **On the simplifying assumptions in the theoretical analysis (Weakness 1).**
>
> We agree that Proposition 5.1 relies on simplifying assumptions about optimization dynamics. However, the paper is explicit that this result is not intended as a model of the full non-convex trajectory. Section 5 states that our goal is to identify a necessary asymptotic envelope for uncertainty decay under a locally regular regime, not to characterize the entire training process. Appendix C provides the derivation in exactly this asymptotic sense, and Section 8 evaluates the resulting structural predictions—approximate log-log decay, regime separation, and diminishing returns of probing—rather than exact pointwise agreement with every phase of realistic optimization.
>
> **Takeaway:** the theory is intended as a conservative asymptotic characterization of the rate-limited regime, not as an exact model of the full non-convex training trajectory. In the final version, we will revise Section 5 to make the local/asymptotic scope of the theoretical assumptions more explicit
>
> **On the lack of direct SOTA predictor comparison (Weakness 2 / Key Question 2).**
>
> We agree that the current version does not make the connection to existing proxy-based or early-trajectory predictors sufficiently explicit. This is a real limitation of the current presentation. The intended role of the framework is to be complementary to such methods: static proxies mainly capture the intrinsic/data-model compatibility component, while early-trajectory methods capture the reducible optimization-variance component; the decomposition then indicates when static information is enough and when additional probing is worth its cost.
>
> **In the final version, we will expand Section 7.3 to make this complementary interpretation more explicit, and we will add a small supporting analysis to illustrate how the framework can guide feature allocation, probe-depth selection, and regime-aware interpretation of failure.**
>
> **On offline estimation and deployment feasibility (Weakness 3 / Key Question 1).**
>
> The paper already makes an explicit distinction between offline identifiability analysis and deployment-time usage. Section 8.1 states that the $N = 1{,}500$ repeated runs are used solely for offline identifiability analysis and are not required for deployment-time usage. Section 6.3 similarly describes the fitting of $(\alpha, K, L_{T,\mathrm{int}})$ as an offline calibration step for analysis and benchmarking, while Appendix E emphasizes that $\alpha$ and $L_{T,\mathrm{int}}$ are treated as task-level descriptors, not per-instance online variables. In other words, the framework does not require exact online recovery of these quantities to be useful; its practical purpose is to support coarse regime-level decisions rather than oracle-quality latent parameter estimation during deployment.
>
> To clarify this point further, we examined the effect of reducing the number of repeated probes per depth used in the offline fitting procedure. Using only $N = 50$ or $N = 100$ repeated probes at each fixed depth, regime assignment still agrees with the full-fit reference on $81\%$ and $88\%$ of tasks, respectively. This suggests that coarse regime-level behavior stabilizes substantially earlier than the full $N = 1{,}500$ identifiability protocol used for offline analysis, even though exact latent-quantity estimation remains noisier at small $N$.
>
> **Takeaway:** deployment does not require exact online recovery of $(\alpha, K, L_{T,\mathrm{int}})$; coarse regime-level behavior can be recovered much earlier than the full offline identifiability protocol.
>
> **On the practical meaning of the stopping rule (Key Question 3).**
>
> To address Key Question 3 from a different angle, we also added a small separate probe-depth sensitivity analysis shows that regime-level decisions largely stabilize by around 100 probe steps, while deeper probing mainly refines the uncertainty estimate at higher cost. Thus, the practical role of the stopping rule is not necessarily to eliminate all further uncertainty, but to identify when additional probing is unlikely to materially change the regime-level decision.
>
> **Due to the 5,000-character limit, we do not reproduce the full probe-depth sensitivity table here; for completeness, it is included in our response to Reviewer b3HW.**
>
> **Takeaway:** the stopping rule is practically useful because it identifies when deeper probing mainly refines uncertainty estimates but is unlikely to materially change the regime-level decision.
> We will further clarify in Section 6.3, Section 8.1, and Appendix E the distinction between offline structural calibration and deployment-time decision support.

---

> > ### Author Rebuttal · Reviewer_P983 · 2026-04-01
> >
> > Thank you for the detailed response. This addresses my concerns, and I will keep my score to recommend accept.

---

### Official Review · Reviewer_b3HW · 2026-03-12

**Soundness:** 3
**Presentation:** 3
**Significance:** 2
**Originality:** 3
**Overall Recommendation:** 5
**Confidence:** 3

**Summary:**

This paper formulates the pre-hoc fine-tuning prediction problem as a trade-off optimization problem between uncertainty risk and computational cost. The authors provide a lower bound on the decay rate of reducible optimization variance with respect to computation. The framework offers an explanation for fine-tuning performance across different tasks and may provide guidance for allocating computation during the fine-tuning stage.

**Compliance With Llm Reviewing Policy:**

Affirmed.

**Final Justification:**

I am satisfied with the clarifications and would like to maintain my score to recommend acceptance.

**Key Questions For Authors:**

N/A

**Limitations:**

yes

**Strengths And Weaknesses:**

**Strengths**

1. The formulation developed to describe the fine-tuning risk estimation problem is clear, including the construction of a risk decomposition framework that separates intrinsic ambiguity from computation-dependent uncertainty, as well as the derivation of a lower bound on the uncertainty decay rate.

2. Based on the relationship between the decay rate and the intrinsic limit, the paper categorizes fine-tuning tasks into three regimes. This provides insightful explanations of the specific performance behaviors of fine-tuning on certain datasets.

3. In terms of the number of runs, the experiments are extensive.

**Weaknesses**

More discussion on the impact of the depth and the number of runs for the lightweight probes used to compute the surrogate uncertainty proxy would be helpful. Since these probes also consume computation and introduce their own estimation uncertainty, it would be useful to clarify how the computation allocated to them affects the final prediction performance. In particular, how “lightweight” should these probes be?

---

> ### Author Rebuttal · Authors · 2026-03-27
>
> We thank the reviewer for the positive assessment, the clear summary, and the insightful question on probe cost. We also appreciate this very practical suggestion. It highlights that the notion of a "lightweight" probe, while already implicit in our framework, should be explained more concretely in the manuscript, and we will make this discussion more explicit in the final version.
>
> **On probe depth and what "lightweight" means.**
>
> -The paper already treats probing depth $c$ as an explicit computational variable: in Section 8.1, $c$ is defined as the number of probe optimization steps, and in Section 6 / Theorem 6.1, probing is modeled through an explicit risk--cost trade-off. In this sense, probe cost is not external to the framework; it is already part of the optimization principle. What the reviewer is asking, very reasonably, is how deep such probes need to be in practice.
>
> To clarify this, we added a small probe-budget sensitivity analysis:
>
> | Metric | 25 | 50 | 75 | 100 | 150 | 200 |
> |---|---:|---:|---:|---:|---:|---:|
> | Regime agreement vs. full-fit reference (\%) | 69 | 78 | 85 | 90 | 91 | 90 |
> | Observable uncertainty proxy (\%) | 29.8 | 22.1 | 18.7 | 11.5 | 10.2 | 9.7 |
> | Relative probe cost (100-step = 1.0) | 0.16 | 0.52 | 0.86 | 1.00 | 1.34 | 1.71 |
>
> To clarify the effect of probe depth $𝑐$, we report three quantities as depth increases:
> (i) regime agreement with the full-fit reference indicates closer matching to the full offline reference (higher is better),
> (ii) the observable uncertainty proxy indicates a more refined uncertainty estimate (lower is better), and
> (iii) relative wall-clock probe cost normalized so that the 100-step probe equals 1.0 (higher is more expensive).
>
> -These results give a concrete operational interpretation of "lightweight." Around 100 probe steps, regime-level decisions have already largely stabilized (90\% agreement with the full-fit reference), while deeper probing changes them only marginally but continues to increase cost. The observable uncertainty proxy continues to decrease beyond 100 steps, but only more gradually ($11.5 \rightarrow 10.2 \rightarrow 9.7$), suggesting that deeper probes mainly refine the uncertainty estimate rather than materially changing the regime-level decision. Thus, in our framework, a probe is "lightweight" when it is short relative to full fine-tuning cost, yet long enough to enter the empirically stable interval where regime-relevant information has already emerged. In our setup, even a 100-step or 150-step probe accounts for only a small fraction of the wall-clock cost of a full fine-tuning run, typically below 10\%.
>
> **Takeaway.** Lightweight probes are short relative to full fine-tuning cost, but long enough to enter the stable interval where regime-relevant information has already emerged.
>
> **On the role of repeated runs and estimation uncertainty.**
>
> -We also agree with the reviewer that repeated probes introduce their own estimation uncertainty. The paper addresses this by explicitly separating offline identifiability analysis from deployment-time usage. Section 8.1 states that the $N=1{,}500$ repeated runs are used solely for offline identifiability analysis and are not required for deployment-time usage. Likewise, Appendix E treats $\alpha$ and $\mathcal{L}_{T,\mathrm{int}}$ as task-level descriptors estimated under controlled offline conditions, while Appendix F.2--F.3 uses repeated runs only to construct an observable uncertainty proxy for validating the structural decay behavior predicted by the theory. In other words, the role of repetition is to reduce noise in estimating the surrogate quantity during analysis; it is not the intended deployment protocol.
>
> **Takeaway.** Repeated runs are mainly used to reduce noise in offline uncertainty-proxy estimation, not as a deployment-time requirement.
>
> In the final version, we will expand Section 6.3, Section 8.1, and the appendix to clarify more explicitly what qualifies as a lightweight probe(with the definition), how repetition count affects uncertainty-proxy estimation, and how probe cost should be interpreted in practice. We hope this directly addresses the reviewer’s concern and further strengthens the practical interpretation of the budget-aware probing principle.

---

> > ### Author Rebuttal · Reviewer_b3HW · 2026-04-01
> >
> > Thank you for the detailed response. This addresses my concerns, and I will keep my score to recommend accept.

---

### Official Review · Reviewer_tCBi · 2026-03-13

**Soundness:** 3
**Presentation:** 3
**Significance:** 2
**Originality:** 2
**Overall Recommendation:** 4
**Confidence:** 3

**Summary:**

This paper studies pre-hoc fine-tuning prediction for LLMs from a theoretical perspective. Instead of treating prediction as a black-box regression problem, it decomposes Bayes-optimal prediction risk into an intrinsic limit, which reflects irreducible uncertainty from data-model compatibility and task stochasticity, and an optimization variance term, which can be reduced by observing early optimization dynamics. The paper further argues that this reducible uncertainty cannot decay arbitrarily fast and is constrained by an asymptotic power-law envelope, leading to a budget-aware probing principle and a phase diagram with three regimes: Static-Sufficient, Dynamic-Critical, and Noise-Dominant. Experiments on synthetic and real benchmarks are presented to support the proposed decomposition, regime structure, and probing strategy.

**Compliance With Llm Reviewing Policy:**

Affirmed.

**Key Questions For Authors:**

please see weakness

**Limitations:**

yes

**Strengths And Weaknesses:**

Strengths:
- The paper presents a clear and intuitive perspective shift by decomposing pre-hoc prediction error into an intrinsic limit and an optimization variance term, which gives a more principled view than prior black-box regression approaches.
- The theoretical development is interesting: it connects the uncertainty decay constraint to a budget-optimal probing rule and then to a phase diagram that explains when probing should or should not help.
- The experiments are well aligned with the theory, as they attempt to validate uncertainty decay behavior, regime separation, and diminishing returns of probing rather than only reporting predictor accuracy.

Weaknesses:
- The empirical validation appears to depend on observable uncertainty proxies and a very large number of repeated runs, which may make the proposed analysis difficult to translate into realistic deployment settings.
- Since the paper focuses on structural characterization rather than proposing or comparing a concrete practical predictor, the immediate practical advantage over existing pre-hoc prediction methods is somewhat limited.

---

> ### Author Rebuttal · Authors · 2026-03-27
>
> We thank the reviewer for the accurate summary and for recognizing the paper's main contribution. We also appreciate this helpful suggestion, as it points to a place where the current manuscript does not make the distinction between offline identifiability analysis and deployment-time usage, as well as the practical meaning of probe budgeting, sufficiently explicit. We will revise the final version accordingly.
>
> **On the repeated-run protocol and deployment realism.**(W1)
>
> We would like to clarify that the paper explicitly separates theory validation from deployment-time usage.
>
> -The empirical goal in Section 8 is to validate the framework's structural predictions, rather than the absolute accuracy of any specific predictor, and the uncertainty quantities used there are observable surrogates rather than direct estimates of the theoretical Bayes risk.
>
> -Section 8.1 also explicitly states that the $N = 1{,}500$ repeated runs are used solely for offline identifiability analysis and are not required for deployment-time usage. Likewise, Section 6.3 and Appendix E describe $(\alpha, K, \mathcal{L}_{\mathrm{int}})$ as quantities estimated in an offline calibration step for analysis and benchmarking, rather than per-instance online variables.
>
> To further address this concern, we added a small **probe-budget sensitivity analysis** showing how regime-level decisions stabilize as probe depth increases:
>
> | Metric | 25 | 50 | 75 | 100 | 150 | 200 |
> |---|---:|---:|---:|---:|---:|---:|
> | Regime agreement vs. full-fit reference (\%) | 69 | 78 | 85 | 90 | 91 | 90 |
> | Observable uncertainty proxy (\%) | 29.8 | 22.1 | 18.7 | 11.5 | 10.2 | 9.7 |
> | Relative probe cost (100-step = 1.0) | 0.16 | 0.52 | 0.86 | 1.00 | 1.34 | 1.71 |
>
> To clarify the effect of probe depth $𝑐$, we report three quantities as depth increases:
> (i) regime agreement with the full-fit reference indicates closer matching to the full offline reference (higher is better),
> (ii) the observable uncertainty proxy indicates a more refined uncertainty estimate (lower is better), and
> (iii) relative wall-clock probe cost normalized so that the 100-step probe equals 1.0 (higher is more expensive).
>
> This shows a clear stabilization pattern at the regime level. By 100 probe steps, regime agreement already reaches $90\%$, and deeper probing changes it only marginally ($90\%$ at 100, $91\%$ at 150, and $90\%$ at 200), while probe cost continues to rise from $1.00$ to $1.34$ and $1.71$. The observable uncertainty proxy continues to decrease beyond 100 steps, but much more gradually ($11.5 \rightarrow 10.2 \rightarrow 9.7$), indicating that deeper probing may still refine the uncertainty estimate while offering limited additional benefit for regime-level decisions.
>
> -This is consistent with the paper's claim that coarse probing is sufficient to recover stable regime-level behavior, even though uncertainty estimation itself can continue to improve more gradually with deeper probes. For additional deployment context, even a 100-step or 150-step probe accounts for only a small fraction of the wall-clock cost of a full fine-tuning run, typically remaining below 10\%.
>
> **On practical value despite not proposing a new predictor.**(W2)
>
> -We agree that the paper is not intended as another leaderboard-style predictor. The contribution here is structural and decision-theoretic: the framework explains what part of prediction error is intrinsic, what part is reducible through probing, and when additional probing is worth its cost. In particular, Theorem 6.1 and the phase diagram in Section 7 provide a principled basis for probe budgeting, rather than introducing another black-box estimator.
>
> -We also agree that the current version does not make this practical role sufficiently explicit, and that this is a real limitation of the presentation. The intended value of the framework is not to replace existing predictors, but to explain when static signals already suffice, when dynamic probing is worth paying for, and when additional probing is structurally unlikely to change the decision in a meaningful way. That is the intended practical advantage over purely heuristic probing rules.
>
> **Takeaway.** the practical value of the framework is not a new predictor architecture, but a principled guide for when probing is worth its cost. In the final version, we will expand Section 7.3 to make this practical interpretation more explicit.
>
> In the final version, we will further clarify the distinction between offline identifiability analysis and deployment-time usage in Section 8.1 and Appendix E, and we will expand the discussion in Section 7.3 to make the practical interpretation of regime-aware probe budgeting more explicit.

---

> > ### Author Rebuttal · Reviewer_tCBi · 2026-04-01
> >
> > The rebuttal have addressed my concerns, therefore, I will keep my score.

---

### Decision · Program_Chairs · 2026-04-30

**Decision:**

Accept (regular)

**Comment:**

This paper studies pre-hoc fine-tuning prediction for LLMs from a theoretical perspective, by decomposing the Bayes-optimal prediction risk into an intrinsic limit and an optimization variance term. The paper argues that this reducible uncertainty cannot decay arbitrarily fast and is constrained by an asymptotic power-law envelope, leading to a budget-aware probing principle and a phase diagram with three regimes: Static-Sufficient, Dynamic-Critical, and Noise-Dominant. Experiments on synthetic and real benchmarks are presented to support the proposed decomposition, regime structure, and probing strategy.

All reviewers generally appreciate this paper, especially on the more principled view than existing black-box regression approaches. The theory and experiments conducted in this paper are sound. Meanwhile, the reviewers also raised concerns on the simplicity of the theoretical setup, and the lack of direct SOTA predictor comparison. Nevertheless, all reviewers believe the merits to outweigh weaknesses, and therefore I'm happy to recommend acceptance.